# *NCBP2* modulates neurodevelopmental defects of the 3q29 deletion in *Drosophila* and *Xenopus laevis* models

**Mayanglambam Dhruba Singh**[1☯], **Matthew Jensen**[1☯], **Micaela Lasser**[2], **Emily Huber**[1], **Tanzeen Yusuff**[1], **Lucilla Pizzo**[1], **Brian Lifschutz**[1], **Inshya Desai**[1], **Alexis Kubina**[1], **Sneha Yennawar**[1], **Sydney Kim**[2], **Janani Iyer**[1], **Diego E. Rincon-Limas**[3], **Laura Anne Lowery**[2,4], **Santhosh Girirajan**[1,5]*

**1** Department of Biochemistry and Molecular Biology, Pennsylvania State University, University Park, Pennsylvania, United States of America, **2** Department of Biology, Boston College, Chestnut Hill, Massachusetts, United States of America, **3** Department of Neurology, McKnight Brain Institute, University of Florida, Gainesville, Florida, United States of America, **4** Department of Medicine, Boston University Medical Center, Boston, Massachusetts, United States of America, **5** Department of Anthropology, Pennsylvania State University, University Park, Pennsylvania, United States of America

☯ These authors contributed equally to this work.
* sxg47@psu.edu

**Data Availability Statement:** Gene expression data for the six *Drosophila* individual and pairwise RNAi knockdown of homologs of 3q29 genes are deposited in the GEO (Gene Expression Omnibus)

## Abstract

The 1.6 Mbp deletion on chromosome 3q29 is associated with a range of neurodevelopmental disorders, including schizophrenia, autism, microcephaly, and intellectual disability. Despite its importance towards neurodevelopment, the role of individual genes, genetic interactions, and disrupted biological mechanisms underlying the deletion have not been thoroughly characterized. Here, we used quantitative methods to assay *Drosophila melanogaster* and *Xenopus laevis* models with tissue-specific individual and pairwise knockdown of 14 homologs of genes within the 3q29 region. We identified developmental, cellular, and neuronal phenotypes for multiple homologs of 3q29 genes, potentially due to altered apoptosis and cell cycle mechanisms during development. Using the fly eye, we screened for 314 pairwise knockdowns of homologs of 3q29 genes and identified 44 interactions between pairs of homologs and 34 interactions with other neurodevelopmental genes. Interestingly, *NCBP2* homologs in *Drosophila* (*Cbp20*) and *X. laevis* (*ncbp2*) enhanced the phenotypes of homologs of the other 3q29 genes, leading to significant increases in apoptosis that disrupted cellular organization and brain morphology. These cellular and neuronal defects were rescued with overexpression of the apoptosis inhibitors *Diap1* and *xiap* in both models, suggesting that apoptosis is one of several potential biological mechanisms disrupted by the deletion. *NCBP2* was also highly connected to other 3q29 genes in a human brain-specific interaction network, providing support for the relevance of our results towards the human deletion. Overall, our study suggests that *NCBP2*-mediated genetic interactions within the 3q29 region disrupt apoptosis and cell cycle mechanisms during development.

database with accession code GSE128094, and the raw RNA Sequencing files are deposited in the SRA (Sequence Read Archive) with BioProject accession PRJNA526450. All other data generated and analyzed in study are included in the manuscript and supporting files.

**Funding:** This work was supported by a Basil O'Connor Award from the March of Dimes Foundation (#5-FY14-66; https://www.marchofdimes.org/), National Institutes of Health R01-GM121907 (https://www.nigms.nih.gov/), a NARSAD Young Investigator Grant from the Brain and Behavior Research Foundation (22535; https://www.bbrfoundation.org/), and resources from the Huck Institutes of the Life Sciences (https://www.huck.psu.edu/) to S.G., National Institutes of Health T32-GM102057 to M.J., and National Institutes of Health R01-MH109651 (https://www.nimh.nih.gov) to L.A.L. The funders had no role in study design, data collection and analysis, decision to publish, or preparation of the manuscript.

**Competing interests:** The authors have declared that no competing interests exist.

## Author summary

Rare copy-number variants, or large deletions and duplications in the genome, are associated with a wide range of neurodevelopmental disorders. The 3q29 deletion confers an increased risk for schizophrenia and autism. To understand the conserved biological mechanisms that are disrupted by this deletion, we systematically tested 14 individual homologs and 314 pairwise interactions of 3q29 genes for neuronal, cellular, and developmental phenotypes in *Drosophila melanogaster* and *Xenopus laevis* models. We found that multiple homologs of genes within the deletion region contribute towards developmental defects, such as larval lethality and disrupted cellular organization. Interestingly, we found that *NCBP2* acts as a key modifier gene within the region, enhancing the developmental phenotypes of each of the homologs for other 3q29 genes and leading to disruptions in apoptosis and cell cycle pathways. Our results suggest that multiple genes within the 3q29 region interact with each other through shared mechanisms and jointly contribute to neurodevelopmental defects.

## Introduction

Rare copy number variants (CNVs), including deletions and duplications in the human genome, significantly contribute to complex neurodevelopmental disorders such as schizophrenia, intellectual disability/developmental delay, autism, and epilepsy [1,2]. Despite extensive phenotypic heterogeneity associated with recently described CNVs [3], certain rare CNVs have been linked to specific neuropsychiatric diagnoses. For example, the 22q11.2 deletion (DiGeorge/velocardiofacial syndrome), the most frequently occurring pathogenic CNV, is found in about 1–2% of individuals with schizophrenia [4,5], and animal models of several genes within the region show neuronal and behavioral phenotypes on their own [6,7]. Similarly, the 1.6 Mbp recurrent deletion on chromosome 3q29, encompassing 21 genes, was initially identified in individuals with a range of neurodevelopmental features, including intellectual disability, microcephaly, craniofacial features, and speech delay [8,9]. Further studies have implicated this deletion as a major risk factor for multiple disorders [10]. In fact, the deletion confers a >40-fold increase in risk for schizophrenia [11,12] as well as a >20-fold increase in risk for autism [13]. More recently, two studies have reported decreases in body and brain sizes as well as a range of behavioral and social defects in mouse models of the entire deletion, mimicking the human developmental phenotypes associated with the deletion [14,15].

Identifying the biological underpinnings of the 3q29 deletion is contingent upon uncovering the conserved molecular mechanisms linking individual genes or combinations of genes within the 3q29 region to the neurodevelopmental phenotypes observed in individuals with the entire deletion. Recent studies have suggested a subset of genes in the 3q29 region as potential candidates for these phenotypes based on their established roles in neuronal development [16,17]. For example, *DLG1* is a scaffolding protein that organizes the synaptic structure at neuromuscular junctions [18], affecting both synaptic density and plasticity during development [19]. However, mouse models of *Dlg1*[+/-] did not recapitulate the behavioral and developmental phenotypes observed in mice with the entire deletion [14], suggesting that haploinsufficiency of *DLG1* by itself does not account for the wide range of phenotypes associated with the deletion. Given that genes within rare pathogenic CNV regions tend to share similar biological functions [20] and interact with each other to contribute towards developmental phenotypes [21,22], it is likely that multiple genes within 3q29 jointly contribute to these phenotypes through shared cellular pathways. Therefore, an approach that integrates

functional analysis of individual genes within the 3q29 deletion and their combinatorial effects on neuronal and cellular phenotypes is necessary to understand the pathways and mechanisms underlying the deletion.

Systematic testing of genes within 3q29 towards developmental and cellular phenotypes requires model systems that are amenable for rapid phenotypic evaluation and allow for testing interactions between multiple dosage-imbalanced genes without affecting the viability of the organism. *Drosophila melanogaster* and *Xenopus laevis* provide such powerful genetic models for studying conserved mechanisms that are altered in neurodevelopmental disorders, with the ability to manipulate gene expression in a tissue-specific manner in *Drosophila* [23] and examine developmental defects in *X. laevis* [24]. Both model systems contain homologs for a majority of disease-causing genes in humans, and show a high degree of conservation in key developmental pathways [23,25–27]. For example, *Drosophila* knockdown models of the candidate schizophrenia gene *DTNBP1* showed dysregulation of synaptic homeostasis and altered glutamatergic and dopaminergic neuron function [28,29], and fly models for *UBE3A*, the gene associated with Angelman syndrome, showed sleep, memory and locomotor defects [30]. Furthermore, *X. laevis* models have been widely used to identify morphological and neuronal defects associated with developmental disorders [24], such as dendritic connectivity defects with overexpression of *MECP2*, the causative gene for Rett syndrome [31]. Thus, *Drosophila* and *X. laevis* models of individual CNV homologs and their interactions would allow for a deeper dissection of the molecular mechanisms disrupted by the deletion, complementing the phenotypes documented in mouse models of the entire deletion [14,15].

Here, we used a mechanistic approach to understand the role of individual homologs of 3q29 genes and their interactions towards the cellular processes underlying the deletion. We systematically characterized developmental, cellular, and nervous system phenotypes for 14 conserved homologs of human 3q29 genes and 314 pairwise interactions using *Drosophila*, and validated these phenotypes using *X. laevis*. We found that multiple homologs of genes within the 3q29 region, including *NCBP2*, *DLG1*, *FBXO45*, *PIGZ*, and *BDH1*, contribute to disruptions in apoptosis and cell cycle pathways, leading to neuronal and developmental defects in both model systems. These defects were further enhanced when each of the homologs were concomitantly knocked down with homologs of *NCBP2* in *Drosophila* (*Cbp20*) and *X. laevis* (*ncbp2*), resulting in increased apoptosis and dysregulation of cell cycle genes. Our results support an oligogenic model for the 3q29 deletion, and implicate specific cellular mechanisms disrupted by genes in the deletion region.

## Results

### Reduced expression of individual homologs of 3q29 genes causes global developmental defects

We used reciprocal BLAST and orthology prediction tools (see Methods) to identify fly homologs for 15 of the 21 genes within the 3q29 deletion region (**Fig 1, S1 Table**). We note that the genes and crosses tested in this study are represented as fly gene names along with the human counterparts at first mention in the text, i.e. *Cbp20* (*NCBP2*), and fly genes with allele names in the figures, i.e. *Cbp20*$^{KK109448}$. We found that the biological functions of these 15 genes were also conserved between *Drosophila* and humans, as 61 of the 69 Gene Ontology (88.4%) annotations for the human genes were also annotated for their respective fly homologs (**S1 File**). For example, *dlg1* (*DLG1*) and *Cbp20* (*NCBP2*) share the same roles in both flies and vertebrates, as a scaffolding protein at the synaptic junction [32] and a member of the RNA cap binding complex [33], respectively. We used RNA interference (RNAi) and the *UAS-GAL4* system to knockdown expression levels of fly homologs of genes within the 3q29 region

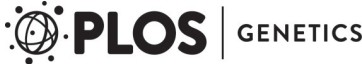

*(Figure: Experimental workflow for Drosophila and Xenopus laevis models of the 3q29 deletion)*

chr. 3

Tested *Drosophila* homologs
Tested *X. laevis* homologs

Human genes: ZDHHC19, OSTalpha, PCYT1A, TCTEX1D2, TM4SF19, UBXN7, TFRC, SMCO1, RNF168, WDR53, FBXO45, LRRC33, CEP19, PIGX, PAK2, SENP5, NCBP2, PIGZ, MFI2, DLG1, BDH1

Fly homologs: app, CG6836, Pcyt2, CG8892, CG5543, Fsn, PIG-X, Pak, Ulp1, Cbp20, PIG-Z, Tsf2, dlg1, CG8888

*Drosophila* single-hit models
Tissue-specific GAL4  ✕  UAS-RNAi

*Drosophila* two-hit models
GMR-GAL4 UAS-RNAi A  ✕  UAS-RNAi B

Global phenotype screening

Genes — Larval lethality, Pupal lethality, Wing defects | Small eyes, Large eyes, Rough eyes | Larval lethality, Pupal lethality, Climbing defect | Lethality, Ectopic veins, Wrinkled wings, Discoloration

*Ubiquitous* *Eye* *Neuronal* *Wing*

Pairwise interaction screening

app, CG6836, Pcyt2, CG8892, CG5543, Fsn, PIG-X, Pak, Ulp1, Cbp20, PIG-Z, dlg1, Tsf2, CG8888

*161 pairwise crosses tested between 3q29 homologs*    *153 pairwise crosses tested with other neurodevelopmental genes*

Neuronal phenotypes

anti-chaoptin

*Climbing ability*    *Axonal targeting*

Cellular phenotypes

DLG    Phalloidin
Necrotic patches

*Adult eye morphology*    *Cellular organization*

Apoptosis and cell cycle mechanisms

DAPI pH3    dcp1    DLG    Gene Ontology terms
Genes

*Cell proliferation*    *Apoptosis*    *Phenotype rescue with apoptosis inhibition*    *Transcriptome analysis*

Validation in *X. laevis* models

GFP+morpholino (2- or 4-cell stage)

*Abnormal eye morphology*    *Rescue with apoptosis inhibition*

GFP+morpholino (2-cell stage)

Tubulin    Uninjected morpholino    Uninjected    morpholino+ *xiap* OE

*Abnormal brain morphology*    *Rescue with apoptosis inhibition*

**Fig 1. Strategy for identifying cellular phenotypes and genetic interactions of homologs of 3q29 genes.** We first knocked down individual or pairs of 14 *Drosophila* homologs of human genes in the 3q29 region using tissue-specific RNAi. After screening for global phenotypes of RNAi lines for individual homologs of 3q29 genes, we tested 314 pairwise gene interactions using the fly eye, and found that *Cbp20* (*NCBP2*) enhanced the phenotypes of other homologs of 3q29 genes and also interacted with homologs of known neurodevelopmental genes outside of the 3q29 region. Next, we assayed for deeper cellular and neuronal phenotypes of flies with individual and pairwise knockdown of homologs of 3q29 genes, and observed cellular defects due to disrupted apoptosis and cell cycle mechanisms. We confirmed our results by rescuing cellular phenotypes with overexpression of the apoptosis inhibitor *Diap1* and by analyzing genes differentially expressed with knockdown of homologs of 3q29 genes. Finally, we tested a subset of three homologs of 3q29 genes in the *X. laevis* vertebrate model system by injecting two- or four-cell stage embryos with GFP and morpholinos (MOs) for *X. laevis* homologs of 3q29 genes to observe abnormal eye morphology, as well as injecting one cell with GFP and MOs at the two-cell stage to observe abnormal brain morphology. We found similar developmental defects in *X. laevis* to those observed in *Drosophila*, including increased apoptosis that was enhanced with pairwise knockdown of homologs of 3q29 genes and rescued with overexpression of the apoptosis inhibitor *xiap*. *X. laevis* embryo diagrams were produced by Nieuwkoop and Faber [117] and adapted from Xenbase [120].

ubiquitously and in neuronal, wing and eye tissues [34] (**Fig 1**). A stock list of the fly lines used in this study and full genotypes for all experiments are provided in **S2 File**. Quantitative PCR (qPCR) confirmed partial knockdown of gene expression for each of the tested homologs (**S2 Table**); fly lines for *CG5359* (*TCTEX1D2*) were excluded from further analysis after additional quality control assessment (see Methods). To identify genes essential for organism survival and neurodevelopment, we first assessed the effect of ubiquitous knockdown of homologs of 3q29 genes using the *da-GAL4* driver (**Fig 2A**). Seven of the 14 homologs, including *dlg1*, *Cbp20*, and *Tsf2* (*MFI2*), showed lethality or severe developmental defects with ubiquitous knockdown, suggesting that multiple homologs of 3q29 genes are essential for viability during early development. Similarly, wing-specific *bx*[MS1096]*-GAL4* knockdown of *Tsf2*, *Cbp20*, *CG8888* (*BDH1*), and *Pak* (*PAK2*) showed severe wing defects, and wing-specific knockdown of *dlg1* showed larval lethality (**S1 Fig**).

Several fly homologs for genes within the 3q29 region have previously been associated with a range of neuronal defects during fly development (**S3 Table**). For example, loss of *dlg1* contributed to morphological and physiological defects at the neuromuscular junction (NMJ), as well as increased brain size, abnormal courtship behavior, and loss of gravitaxis response [35–37]. Similarly, *Pak* mutant flies exhibited extensive defects in the axonal targeting of sensory and motor neurons [38,39], in addition to abnormal NMJ and mushroom body development [40,41]. We sought to determine whether fly homologs for other genes in the 3q29 region also contribute to defects in neuronal function, and therefore performed climbing assays for motor defects and staining of larval brains for axonal targeting with pan-neuronal knockdown of the fly homologs. Interestingly, *Elav-GAL4* mediated pan-neuronal knockdown caused larval or pupal lethality in *dlg1*, *Tsf2*, and *CG5543* (*WDR53*) flies (**Fig 2A**), and about 30% of adult flies with knockdown of *dlg1* did not survive beyond day 5 (**S1 Fig**), indicating an essential role for these genes in neuronal development. Furthermore, we found that flies with pan-neuronal knockdown of several homologs of 3q29 genes, including *dlg1* and *Cbp20*, exhibited a strong reduction in climbing ability over ten days (**Fig 2B, S1 Video**), suggesting that these genes could contribute to abnormalities in synaptic and motor functions [42]. We next examined the axonal projections of photoreceptor cells into the optic lobe by staining third instar larval brains with anti-chaoptin. We found that *GMR-GAL4* mediated eye-specific knockdown of *Cbp20*, *dlg1*, *Pak* and *Fsn* (*FBXO45*) showed several axonal targeting defects (**S1 Fig, S4 Table**). Our results recapitulated the previous findings in *Pak* mutant flies [38], and were similar to targeting defects observed in models of other candidate neurodevelopmental genes, including *Drosophila* homologs for human *DISC1* and *FMR1* [43,44]. Overall, our data show that multiple conserved homologs of genes in the 3q29 region beyond just *dlg1* or *Pak* are important for *Drosophila* neurodevelopment.

Fig 2. Neurodevelopmental defects in flies with knockdown of individual homologs of 3q29 genes. (**A**) Percentage of flies with tissue-specific RNAi knockdown of homologs of 3q29 genes (listed with their human counterparts) that manifest lethality or developmental phenotypes. (**B**) Eight homologs of 3q29 genes with pan-neuronal RNAi knockdown showed defects in climbing ability over ten days (two-way repeated measures ANOVA, $p<1\times10^{-4}$, df = 8, F = 21.097). Data represented show mean ± standard deviation of 10 independent groups of 10 flies for each homolog. (**C**) Representative brightfield adult eye images of flies with eye-specific *GMR-GAL4;UAS-Dicer2* (scale bar = 100 μm) RNAi knockdown of individual homologs of 3q29 genes show rough eye phenotypes. The boxplot shows *Flynotyper*-derived phenotypic scores for eyes with knockdown of homologs of 3q29 genes (n = 10–14, *p < 0.05, one-tailed Mann–Whitney test with Benjamini-Hochberg correction). (**D**) Boxplot of adult eye area in flies with *GMR-GAL4* RNAi knockdown of fly homologs of 3q29 genes (n = 13–16, *p < 0.05, two-tailed Mann–Whitney test with Benjamini-Hochberg correction). (**E**) Confocal images of pupal eyes (scale bar = 5 μm) stained with anti-DLG (top) and larval eye discs (scale bar = 30 μm) stained with anti-pH3 (middle) and anti-dcp1 (bottom) illustrate cellular defects posterior to the morphogenetic furrow (white box) upon knockdown of select fly homologs of 3q29 genes. Yellow circles in DLG images indicate cone cell defects, white circles indicate bristle cell defects, yellow arrows indicate rotation defects, and yellow arrowheads indicate secondary cell defects. We note that pupal eye images were taken at a higher intensity for lines with knockdown of *dlg1* to account for reduced expression of DLG (see Methods), as these images were only for visualization of cell boundaries in the pupal eye and not for any quantitative analysis. (**F**) Boxplot of pH3-positive cells in larval eye discs of flies with knockdown of homologs of 3q29 genes (n = 9–12, *p < 0.05, two-tailed Mann–Whitney test with Benjamini-Hochberg correction). (**G**) Boxplot of dcp1-positive cells in larval eye discs of flies with knockdown of homologs of 3q29 genes (n = 11–12, *p < 0.05, two-tailed Mann–Whitney test with Benjamini-Hochberg correction). All boxplots indicate median (center line), 25th and 75th percentiles (bounds of box), and minimum and maximum (whiskers), with red dotted lines representing the control median. Results for a subset of climbing ability, adult eye area, and pH3 staining experiments were replicated in independent experimental batches (S14 Fig). A list of full genotypes for fly crosses used in these experiments is provided in S2 File.

## *Drosophila* eye models for genes within the 3q29 region show cellular defects

The *Drosophila* compound eye has been classically used to perform high-throughput genetic screens and quantitative assays of cellular and neurodevelopmental defects [45]. In fact, about two-thirds of all vital genes in the fly genome are predicted to be involved in fly eye development [46]. For instance, the *Drosophila* eye model was recently used to screen a large set of intellectual disability genes [47], and genetic interaction studies using the fly eye have identified modifier genes for Rett syndrome, spinocerebellar ataxia type 3, and other conserved developmental processes [48–50]. We used the developing fly eye as an *in vivo* system to quantify the effect of gene knockdown on adult eye morphology, cellular organization in the pupal eye, and cell proliferation and death in the larval imaginal eye disc (**Fig 1**, **S2 Fig**). The wild-type adult *Drosophila* eye consists of about 750 ommatidia containing different cell types arranged in a regular hexagonal structure, which can be easily perturbed by genetic modifications [51,52]. Because of this, we first performed eye-specific RNAi knockdown of fly homologs of genes in the 3q29 region using *GMR-GAL4*, and measured the rough eye phenotype of each knockdown line using *Flynotyper*, a quantitative tool that calculates a phenotypic score based on defects in ommatidial arrangement [53]. We found that eye-specific knockdown of 8/13 homologs of 3q29 genes showed significant external eye phenotypes compared with control *GMR-GAL4* flies, while knockdown of *Tsf2* caused lethality (**Fig 2C**, **S3 Fig**). For example, knockdown of *Cbp20* resulted in a severe rough eye phenotype that was comparable to knockdown of other neurodevelopmental genes [53], such as *Prosap* (*SHANK3*) and *kis* (*CHD8*) (**S5 Table**).

To examine the cellular mechanisms underlying the rough eye phenotypes observed with knockdown of fly homologs of 3q29 genes, we first measured changes in area and ommatidial size of the adult eyes. We found a significant reduction in eye size with knockdown of *CG8888* and *Cbp20*, while the eyes of flies with knockdown of *dlg1* were significantly larger than *GMR-GAL4* controls (**Fig 2D**). Similarly, we observed decreases in ommatidial diameter with knockdown of *Cbp20* and *CG8888*, suggesting that these genes may also contribute to abnormal cell growth phenotypes (**S3 Fig**). We also assessed the cellular structure of 44 hour-old pupal eyes by staining the ommatidial and photoreceptor cells with anti-DLG, a septate junction marker, and Phalloidin, a marker for F-actin at cell boundaries (**S2 Fig**). We found that knockdown of 11/12 tested homologs of 3q29 genes caused disorganization or loss of the photoreceptor neurons and ommatidial cells (**Fig 2E**, **S4 Fig**, **S6 Table**). For example, pupal eyes with knockdown of *CG8888*, *dlg1*, *Cbp20* and *CG5543* all showed defects in cone cell orientation and ommatidial rotation compared with control *GMR-GAL4* flies. Furthermore, *Cbp20* and *dlg1* knockdown flies showed hexagonal defects and severe disorganization of photoreceptor neurons, while *Cbp20* knockdown flies also showed fused secondary cells and *dlg1* knockdown flies showed a complete loss of bristle cells.

We next hypothesized that abnormal proliferation and apoptosis could contribute to the cellular defects observed with knockdown of fly homologs of 3q29 genes. To test this, we stained the third instar larval eye discs for select knockdowns of individual homologs of 3q29 genes with anti-pH3 (phospho-Histone H3 (Ser10)) and *Drosophila* caspase-1 (dcp1), markers for proliferating and apoptotic cells, and quantified the number of cells posterior to the morphogenetic furrow (**S2 Fig**). We observed a significant decrease in pH3-positive cells for *CG8888* knockdown flies and trends towards increased pH3-positive cells for *PIG-Z* (*PIGZ*) knockdown flies compared with *GMR-GAL4* controls (p = 0.165) (**Fig 2F**, **S4 Fig**), while knockdown of *dlg1* led to significant increases in cells stained with bromodeoxyuridine (BrdU), a marker for replicating cells (**S4 Fig**). Flies with knockdown of *Cbp20 or dlg1* also

**Table 1. Summary of major experiments for knockdown of homologs of 3q29 genes show widespread cellular and neuronal defects.**

| Experiment | | RNAi knockdown of *Drosophila* homologs of 3q29 genes | | | | | | |
|---|---|---|---|---|---|---|---|---|
| **Phenotype** | **Assay** | *Cbp20* | *dlg1* | *Cbp20/dlg1* | *Cbp20/Fsn* | *Cbp20/CG8888* | *Cbp20/ Diap1* | *dlg1/ Diap1* |
| Adult eye morphology | Rough eye phenotype | Rough eye | Rough eye | Enhanced rough eye | Enhanced rough eye | Enhanced rough eye | Rescue | Rescue |
| | Necrotic patches | None (Present in homozygous KD) | None | Yes (more severe in homozygous KD) | Yes | None | None | None |
| | Eye area | Decreased area | Increased area | NA | NA | NA | Rescue | Rescue |
| Neuronal phenotypes | Climbing ability | Climbing defects | Climbing defects | Enhanced climbing defects | Enhanced climbing defects | NA | NA | NA |
| | Axonal targeting | Axon targeting defects | Axon targeting defects | Enhanced targeting defects | Enhanced targeting defects | NA | Rescue | Rescue |
| Cell organization (pupal eye) | DLG staining | Cellular defects | Cellular defects | Enhanced cellular defects | Enhanced cellular defects | Enhanced cellular defects | Rescue | Rescue |
| | Phalloidin staining | Loss of photoreceptors | Loss of photoreceptors | No change | Enhanced photoreceptor loss | Enhanced photoreceptor loss | Rescue | Rescue |
| Cell cycle (larval eye disc) | pH3 staining | No change | No change | No change | No change | Decreased proliferation | NA | NA |
| | BrdU staining | No change | Increased proliferation | NA | NA | NA | NA | NA |
| Apoptosis (larval eye disc) | dcp1 staining | Increased apoptosis | Increased apoptosis | Increased apoptosis | Increased apoptosis | Increased apoptosis | Rescue | Rescue |
| | TUNEL assay | Increased apoptosis | Increased apoptosis | Increased apoptosis | Increased apoptosis | Increased apoptosis | Rescue | Rescue |
| Cellular phenotypes (larval wing disc) | pH3 staining | Decreased proliferation | Increased proliferation | NA | NA | NA | NA | NA |
| | dcp1 staining | Increased apoptosis | Increased apoptosis | NA | NA | NA | NA | NA |
| RNA sequencing (adult heads) | Differential gene expression | Synaptic transmission, metabolism | Synaptic transmission, ion transport | Cellular respiration, protein folding | Cell cycle, response to stimulus | NA | NA | NA |
| **Experiment** | | Morpholino knockdown of *X. laevis* homologs of 3q29 genes | | | | | | |
| **Phenotype** | **Assay** | *ncbp2* | *fbxo45* | *pak2* | *ncbp2/fbxo45* | *ncbp2/pak2* | *ncbp2/ xiap* | *fbxo45/ xiap* |
| Craniofacial morphology | Eye area | Decreased area | Decreased area | Decreased area | NA | NA | Rescue | NA |
| | Midbrain area | Decreased area | Decreased area | Decreased area | No change | No change | Rescue | NA |
| | Forebrain area | Decreased area | Decreased area | Decreased area | Decreased area | No change | Rescue | NA |
| Apoptosis | Cleaved caspase-3 levels | Increased caspase-3 | Increased caspase-3 | NA | Increased caspase-3 | NA | Rescue | Rescue |

showed a significant increase in apoptotic dcp1-positive cells compared with controls (**Fig 2G**), which we validated using TUNEL assays for these lines (**S4 Fig**). We further tested for proliferation and apoptosis in the third instar larval wing discs of flies with knockdown of homologs of 3q29 genes using the $bx^{MS1096}$-GAL4 driver, and observed changes in both processes with knockdown of *dlg1*, *CG8888* and *Cbp20* (**S5 Fig**). Knockdown of *Cbp20* in particular showed dcp1-positive staining across the entire wing pouch in the larval wing disc. These data suggest that knockdown of multiple fly homologs of genes in the 3q29 region contribute to defects in apoptosis and proliferation during early development, leading to the observed defects in cell count and organization (**Table 1**).

## Interactions between fly homologs of 3q29 genes enhance neuronal phenotypes

As knockdown fly models for homologs of multiple 3q29 genes showed a variety of neuronal, developmental, and cellular defects, we hypothesized that these genes could interact with each other to further disrupt cellular processes during development. We therefore generated *GMR-GAL4* recombined lines for nine fly homologs of 3q29 genes, and crossed these lines with multiple RNAi or mutant lines for other homologs to generate 161 two-hit crosses for testing 94 pairwise gene interactions (**Fig 1**, **S7 Table**). We found a significant enhancement in eye phenotypic severity, measured using *Flynotyper* and validated with a second line when available, for 39 pairwise knockdowns compared with recombined lines crossed with control flies (represented in the figures as *Cbp20^{KK109448}/Control*) (**Fig 3A**, **S6 Fig**, **S7 Fig**). In fact, we found that 19 out of 21 pairwise interactions involving *Cbp20* as either a first or second-hit gene resulted in more severe eye phenotypes, suggesting that reduced expression of *Cbp20* drastically modifies the morphological phenotypes of other homologs of 3q29 genes (**Fig 3B–3D**). For further validation, we compared pairs of reciprocal crosses (i.e. *Fsn/CG8888* versus *CG8888/Fsn*) and confirmed concordant results for 19/26 reciprocal interactions, including 14/16 reciprocal interactions involving *Cbp20* (**S7 Table**). We also found a non-significant increase in severity for *dlg1/Pak* knockdown flies using both RNAi and mutant lines, concordant with enhanced neuromuscular junction and circadian rhythm defects observed in mutant *dlg1/Pak* flies described by Grice and colleagues [54].

As *Cbp20* knockdown enhanced the rough eye phenotypes of multiple other homologs, we next tested for enhancement of neuronal defects among flies with knockdown of *Cbp20* and homologs of other 3q29 genes. We found that simultaneous knockdown of *Cbp20* with *dlg1* or *Fsn* led to an increase in severity of axon targeting defects (**Fig 3E**). For instance, while knockdown of *Cbp20* mostly led to mild-to-moderate axon targeting defects, such as loss of R7-R8 axon projection into the medulla, we observed more severe loss of projection for all axons with simultaneous knockdown of *Cbp20* and *dlg1* or *Fsn* (**S4 Table**). We also tested pan-neuronal *Elav-GAL4* knockdown of select pairs of homologs, and found that both *Cbp20/dlg1* and *Cbp20/Fsn* significantly enhanced the climbing defects observed with knockdown of *Cbp20* alone (**Fig 3F**, **S2 Video**). Overall, these data show that *Cbp20* interacts with other homologs of genes in the 3q29 region to enhance the observed cellular and neuronal defects (**Table 1**).

To further characterize the functional effects of interactions between homologs of 3q29 genes, we analyzed changes in gene expression by performing RNA-sequencing of heads from flies with select pan-neuronal knockdown of individual (*Cbp20*, *dlg1*, *Fsn*, and *Pak*) and pairs (*Cbp20/dlg1* and *Cbp20/Fsn*) of homologs of 3q29 genes. We identified differentially-expressed genes in each of the tested fly models compared with *Elav-GAL4* controls, and performed enrichment analysis on both differentially-expressed fly genes and their corresponding human homologs (**S3 File**). We found that knockdown of each individual homolog showed enrichment for dysregulation of cellular and developmental processes (**S8 Fig**). For example, flies with knockdown of *dlg1* and *Cbp20* showed enrichment for dysregulation of homologs for human synaptic transmission genes, such as *Glt* (*NLGN1*) and *nAChRβ3* (*HTR3A*). Furthermore, flies with knockdown of *Cbp20* were enriched for dysregulated fly genes related to metabolic processes, while knockdown of *Fsn* led to dysregulation of fly genes involved in response to external stimuli and immune response. We also found that homologs of key signaling genes dysregulated in mouse models of the 3q29 deletion, reported by Baba and colleagues [15], were differentially expressed in our fly models for homologs of 3q29 genes. In fact, knockdown of *Fsn* led to altered expression for each of the "early immediate" signaling genes dysregulated in the deletion mouse model [15]. While dysregulated genes in *Cbp20/dlg1* knockdown flies showed enrichments for protein folding and sensory perception, *Cbp20/Fsn* knockdown flies

**Fig 3. Screening for pairwise interactions of fly homologs of 3q29 genes in the *Drosophila* eye and nervous system.** (**A**) Heatmap showing average changes in phenotypic scores for pairwise *GMR-GAL4* RNAi knockdown of fly homologs of 3q29 genes in the adult eye, compared with recombined lines for individual homologs of 3q29 genes crossed with controls. Gray boxes indicate crosses without available data. Boxplots of phenotypic scores for pairwise knockdown of (**B**) *Cbp20* and (**C**) *dlg1* with other fly homologs of 3q29 genes are shown (n = 5–14, *p < 0.05, two-tailed Mann–Whitney test with Benjamini-Hochberg correction). Green arrows indicate an example pair of reciprocal lines showing enhanced phenotypes compared with their respective single-hit recombined controls. Crosses with the mutant line *Tsf2^KG01571^* are included along with RNAi lines for other homologs of 3q29 genes, as eye-specific RNAi knockdown of *Tsf2* was lethal. (**D**) Representative brightfield adult eye images of flies with pairwise knockdown of fly homologs of 3q29 genes (scale bar = 100 μm) show enhancement (Enh.) of rough eye phenotypes compared with recombined lines for individual homologs of 3q29 genes crossed with controls. (**E**) Representative confocal images of larval eye discs stained with anti-chaoptin (scale bar = 30 μm) illustrate enhanced defects (Enh.) in axon targeting (white arrows) from the retina to the optic lobes of the brain with eye-specific knockdown of *Cbp20*/*dlg1* and *Cbp20*/*Fsn* compared with *Cbp20* knockdown. Note that n = 9–17 larval eye disc preparations were assessed for each tested interaction. (**F**) Flies with pan-neuronal *Elav-GAL4* pairwise knockdown of homologs of 3q29 genes showed enhanced defects in climbing ability over ten days (two-way repeated measures ANOVA, p<4.00×10⁻⁴, df = 2, F = 7.966) compared with recombined *Cbp20* knockdown crossed with control. Data represented show mean ± standard deviation of 10 independent groups of 10 flies for each line tested. Results for the climbing assays were replicated in an independent experimental batch (S14 Fig). All boxplots indicate median (center line), 25th and 75th percentiles (bounds of box), and minimum and maximum (whiskers), with red dotted lines representing the control median. A list of full genotypes for fly crosses used in these experiments is provided in S2 File.

were uniquely enriched for dysregulated homologs of cell cycle genes, including *Aura* (*AURKA*), *Cdk1* (*CDK1*), *lok* (*CHEK2*), and *CycE* (*CCNE1*) (S8 Fig). We similarly found 17 differentially-expressed homologs corresponding to human apoptosis genes in *Cbp20*/*Fsn* knockdown flies, including the DNA fragmentation gene *Sid* (*ENDOG*) and the apoptosis

signaling genes *tor* (*RET*) and *Hsp70Bb* (*HSPA1A*). Furthermore, we found a strong enrichment for fly genes whose human homologs are preferentially expressed in early and mid-fetal brain tissues among the dysregulated genes in *Cbp20/Fsn* knockdown flies (**S8 Fig**). These data suggest that *Cbp20* interacts with other homologs of genes in the 3q29 region to disrupt a variety of key biological functions, including apoptosis and cell cycle pathways as well as synaptic transmission and metabolic pathways (**Table 1**).

Finally, to complement the interactions among homologs of 3q29 genes that we identified in *Drosophila*, we examined the connectivity patterns of 3q29 genes within the context of human gene interaction databases. Gene interaction networks derived from co-expression and protein-protein interaction data [55,56] showed large modules of connected genes within the 3q29 region, including a strongly-connected component involving 11/21 3q29 genes (**Fig 4A and 4B**). However, the average connectivity among 3q29 genes within a brain-specific interaction network [57] was not significantly different from the connectivity of randomly-selected sets of genes throughout the genome (**Fig 4C**), suggesting that a subset of genes drive the complexity of genetic interactions within the region. This paradigm was previously observed among genes in the 22q11.2 deletion region, where interactions between *PRODH* and *COMT* modulate neurotransmitter function independently of other genes in the region [58]. In fact, five genes in the 3q29 region, including *NCBP2*, *PAK2*, and *DLG1*, showed significantly higher connectivity to other 3q29 genes compared with the average connectivity of random sets of genes (**Fig 4D**). Interestingly, *NCBP2* showed the highest connectivity of all genes in the region, further highlighting its role as a key modulator of genes within the region.

## Interactions between *Cbp20* and other homologs of 3q29 genes enhance apoptosis defects

Cell death and proliferation are two antagonistic forces that maintain an appropriate number of neurons during development [59]. In fact, both processes have been previously identified as candidate mechanisms for several neurodevelopmental disorders [60–62]. While knockdown of *Cbp20* with other homologs of 3q29 genes likely disrupts multiple cellular processes that contribute towards the enhanced cellular defects, we next specifically investigated the role of apoptosis towards these defects, as larval eye and wing discs with knockdown of *Cbp20* showed strong increases in apoptosis. We observed black necrotic patches on the ommatidia in adult eyes with knockdown of *Cbp20/dlg1* and *Cbp20/Fsn*, indicating that an increase in cell death occurs with these interactions (**Fig 5A**, **S9 Fig**). In fact, significantly larger regions of necrotic patches were observed in flies homozygous for *Cbp20* RNAi and heterozygous for *dlg1* RNAi (see **S2 File** for full genotype annotation), suggesting that the knockdown of both homologs contributes to ommatidial cell death (**Fig 5A**). Furthermore, we found an enhanced disruption of ommatidial cell organization and loss of photoreceptors in pupal flies with concomitant knockdown of *Cbp20* with *dlg1*, *Fsn* or *CG8888*, emphasizing the role of these genes in maintaining cell count and organization (**Fig 5B and 5C**, **S9 Fig**, **S8 Table**). Based on these observations, we next assayed for apoptotic cells in the larval eye discs of flies with knockdown of *Cbp20* and other homologs of 3q29 genes. We observed significant increases in the number of apoptotic cells, as measured by dcp1 (**Fig 5D and 5E**) and TUNEL staining (**S9 Fig**), when *Cbp20* was knocked down along with *CG8888*, *dlg1*, or *Fsn*. *Cbp20/CG8888* knockdown flies also showed a decreased number of pH3-positive cells, suggesting that both apoptosis and proliferation are affected by the interaction between these two genes (**Fig 5F**).

To validate apoptosis as a candidate mechanism for the cellular defects of flies with knockdown of homologs of 3q29 genes, we crossed recombined fly lines for *Cbp20* and *dlg1* with flies overexpressing *Diap1* (death-associated inhibitor of apoptosis). *Diap1* is an E3 ubiquitin

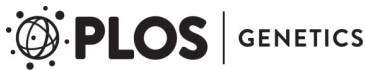

**Fig 4. Connectivity of 3q29 genes in human gene interaction databases.** **(A)** Genetic interactions of 3q29 genes in the context of a general human gene interaction network (GeneMania). The strongly connected component includes 11/21 total 3q29 genes. Black-shaded nodes represent the input 3q29 genes, while grey nodes represent connector genes in the network. Edge color represents the interaction data source (purple: co-expression, orange: predicted interaction), while edge thickness represents weighted scores for each interaction. **(B)** Genetic interactions of 19 genes in the 3q29 region in the context of a brain-specific human gene interaction network (GIANT). Large nodes represent the input 3q29 genes, while small nodes represent connector genes in the network. Edge color represents the weighted score for each interaction, from low connectivity (green) to high connectivity (red). **(C)** Histograms and smoothed normal distributions showing the average connectivity among genes in the 3q29 region (blue) along with two other large CNVs, 16p11.2 (red) and 22q11.2 deletion (green), within a brain-specific gene interaction network. Average connectivity is measured as the shortest weighted distance between two genes, with lower distances representing higher connectivity. Genes within the 3q29 and 22q11.2 deletions were not significantly more connected to each other (p>0.05, one-tailed Mann-Whitney test with Benjamini-Hochberg correction) than random sets of 21 genes throughout the genome (grey). However, genes within the 16p11.2 region were significantly more connected to each other than the random gene sets (p = 0.003, one-tailed Mann-Whitney test with Benjamini-Hochberg correction). **(D)** Pairwise connectivity of individual 3q29 genes within a brain-specific gene interaction network, excluding six genes not present in the network (*RNF168*, *ZDHHC19*, *LRRC33*, *OSTalpha*, *SMCO1*, and *TCTEX1D2*). Average connectivity is measured as the shortest weighted distance between two genes, with lower values representing higher connectivity. Underlined genes have a higher average connectivity (p<0.05, one-tailed Mann-Whitney test with Benjamini-Hochberg correction) to other genes in the region compared with random sets of 21 genes throughout the genome.

ligase that targets *Dronc*, the fly homolog of caspase-9, and prevents the subsequent activation of downstream caspases that lead to apoptosis [63]. We found that overexpression of *Diap1* rescued the adult rough eye phenotypes (**Fig 6A and 6B**, **S10 Fig**) and increased the eye sizes of *Cbp20* and *dlg1* flies (**S10 Fig**). These observations were corroborated by the reversal of

**A** Necrotic patches in the adult eye

Cbp20^{KK109448}/ Control

Cbp20^{KK109448}/ dlg1^{GD4689}

Necrotic patches

Enh.

Homozyg. Cbp20^{KK109448}

Homozyg. Cbp20^{KK109448}/ Heterozyg. dlg1^{GD4689}

Enh.

**B** Cellular organization defects

Cbp20^{KK109448}

Cbp20^{KK109448}/ CG8888^{GD3777}

DLG

Cbp20^{KK109448}/ dlg1^{GD4689}

Cbp20^{KK109448}/ Fsn^{GD11383}

**C** Photoreceptor cell defects

Cbp20^{KK109448}

Cbp20^{KK109448}/ CG8888^{GD3777}

Phalloidin

Cbp20^{KK109448}/ dlg1^{GD4689}

Cbp20^{KK109448}/ Fsn^{GD11383}

**D** Cellular phenotypes of pairwise knockdowns in the larval eye disc

Cbp20^{KK109448}/Control

Cbp20^{KK109448}/ CG8888^{GD3777}

Cbp20^{KK109448}/ dlg1^{GD4689}

Cbp20^{KK109448}/ Fsn^{GD11383}

dcp1

Apoptosis

pH3 DAPI

Proliferation

**E** Apoptosis defects

Fly RNAi lines

**F** Proliferation defects

Fly RNAi lines

**Fig 5. Cellular phenotypes with pairwise knockdown of fly homologs of 3q29 genes.** (**A**) Representative brightfield adult eye images (scale bar = 100 μm) show that heterozygous *GMR-GAL4* RNAi knockdown of *dlg1* enhanced the rough eye phenotype and necrotic patches (yellow circles) of flies heterozygous or homozygous for *Cbp20* RNAi. (**B**) Representative confocal images of pupal eyes (scale bar = 5 μm) stained with anti-DLG illustrate enhanced defects in ommatidial organization upon concomitant knockdown of *Cbp20* with other fly homologs of 3q29 genes compared with *Cbp20* knockdown. Yellow circles in DLG images indicate cone cell defects, white circles indicate bristle cell defects, yellow arrows indicate rotation defects, and yellow arrowheads indicate secondary cell defects. We note that pupal eye images were taken at a higher intensity for lines with knockdown of *Cbp20/dlg1* to account for reduced expression of DLG (see Methods), as these images were only for visualization of cell boundaries in the pupal eye and not for any quantitative analysis. (**C**) Representative confocal images of pupal eyes (scale bar = 5 μm) stained with Phalloidin illustrate enhanced defects in photoreceptor cell count and organization upon concomitant knockdown of *Cbp20* and other fly homologs of 3q29 genes compared with *Cbp20* knockdown. (**D**) Representative confocal images of larval eye discs (scale bar = 30 μm) stained with anti-dcp1 (top) and anti-pH3 (bottom) show enhanced defects in apoptosis and cell proliferation with pairwise knockdown of *Cbp20* and other fly homologs of 3q29 genes compared with recombined *Cbp20* knockdown crossed with controls. (**E**) Boxplot of dcp1-positive cells in the larval eye discs of flies with pairwise knockdown of homologs of 3q29 genes (n = 10–11, *p < 0.05, two-tailed Mann–Whitney test with Benjamini-Hochberg correction). (**F**) Boxplot of pH3-positive cells in the larval eye discs of flies with pairwise knockdown of homologs of 3q29 genes (n = 10–12, *p < 0.05, two-tailed Mann–Whitney test with Benjamini-Hochberg correction). All boxplots indicate median (center line), 25th and 75th percentiles (bounds of box), and minimum and maximum (whiskers), with red dotted lines representing the control median. A list of full genotypes for fly crosses used in these experiments is provided in **S2 File**.

cellular changes in the eye upon *Diap1* overexpression, including the rescue of ommatidial structure and cell count deficits observed with knockdown of *Cbp20* and *dlg1* (**Fig 6D, S10 Fig**). Furthermore, overexpression of *Diap1* led to significant reductions in the number of TUNEL and dcp1-positive cells in the larval eye discs of flies with knockdown of *Cbp20* and *dlg1*, confirming the rescue of apoptosis defects in these flies (**Fig 6E and 6F, S10 Fig**). Interestingly, *Diap1* overexpression also suppressed the photoreceptor axon targeting defects observed with knockdown of *Cbp20* (**Fig 6G, S4 Table**), suggesting that the neuronal defects observed in these flies could be attributed to increased apoptosis. We further confirmed these mechanistic findings by observing increased severity in cellular phenotypes upon overexpression of *Dronc* in *Cbp20* and *dlg1* knockdown flies. For example, we observed black necrotic patches (**Fig 6A and 6C**) and exaggerated apoptotic responses (**Fig 6E and 6F, S10 Fig**) in *Cbp20* knockdown flies with overexpression of *Dronc*. These results suggest that apoptosis mediates the cellular defects observed in flies with knockdown of *Cbp20* and *dlg1*.

### Homologs of 3q29 genes interact with canonical neurodevelopmental genes

We further explored the role of 3q29 genes in neurodevelopmental pathways by screening four fly homologs with strong neurodevelopmental phenotypes (*Cbp20*, *dlg1*, *CG8888*, and *Pak*) for interactions with homologs of 15 known human neurodevelopmental genes, for a total of 60 pairwise interactions and 153 two-hit crosses (**Fig 7A**). We selected these neurodevelopmental genes for screening based on their association with developmental disorders in humans [53,64], and included eight genes associated with apoptosis or cell cycle functions as well as four genes associated with microcephaly [65], a key phenotype observed in approximately 50% of 3q29 deletion carriers [8]. We found that 34 pairwise interactions, validated with a second line when available, led to significant increases in eye phenotypes compared with recombined lines for individual homologs of 3q29 genes (**S9 Table, S11 Fig**). These interactions included 19 validated interactions of homologs of 3q29 genes with apoptosis or cell cycle genes as well as ten interactions with microcephaly genes. We found that 13/15 homologs of neurodevelopmental genes, including all four microcephaly genes, enhanced the phenotypes observed with knockdown of *Cbp20* alone. Furthermore, knockdown of *dlg1* significantly enhanced the ommatidial necrotic patches observed with knockdown of *arm* (*CTNNB1*), while flies with concomitant knockdown of *Cbp20* and *arm* also showed increased necrotic patches (**Fig 7B, S9 Fig**). Interestingly, we also found that knockdown of *CG8888* and *dlg1* suppressed the rough eye phenotypes observed with knockdown of *Prosap* (*SHANK3*), while knockdown of *Pak* suppressed the phenotypes of both *Prosap* and *Pten* (*PTEN*) knockdown flies (**Fig 7B**). Several of these interactions have been previously observed to modulate neuronal function in model systems. For example, *SHANK3* interacts with *DLG1* through the mediator protein DLGAP1 to influence post-synaptic density in mice [66] and binds to proteins in the Rac1 complex, including PAK2, to regulate synaptic structure [67,68]. These results suggest that homologs of 3q29 genes interact with key developmental genes in conserved pathways to modify cellular phenotypes.

### Reduction of 3q29 gene expression causes developmental defects in *Xenopus laevis*

After identifying a wide range of neurodevelopmental defects due to knockdown of fly homologs of 3q29 genes, we sought to gain further insight into the conserved functions of these genes in vertebrate embryonic brain development using the *Xenopus laevis* model system. We examined the effect of targeted knockdown of *ncbp2*, *fbxo45*, and *pak2*, as homologs of these genes displayed multiple severe phenotypes with reduced gene expression in flies. Knockdown

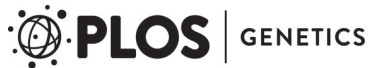

**Fig 6. Rescue of cellular phenotypes due to knockdown of fly homologs of 3q29 genes with overexpression of the apoptosis inhibitor *Diap1*.** (**A**) Representative brightfield adult eye images (scale bar = 100 μm) show rescue of rough eye phenotypes for flies with concomitant *GMR-GAL4* RNAi knockdown of *Cbp20* or *dlg1* and overexpression of *Diap1*, as well as enhanced (Enh.) phenotypes with overexpression of caspase-9 homolog *Dronc*. (**B**) Boxplot of phenotypic scores for flies with knockdown of *Cbp20* or *dlg1* and overexpression of *Diap1* or *Dronc* (n = 8–9, *p < 0.05, two-tailed Mann–Whitney test with Benjamini-Hochberg correction) is shown. (**C**) Box plot showing area of necrotic patches in adult fly eyes with knockdown of *Cbp20* and overexpression of *Dronc* (n = 9, *p = 1.14×10⁻⁴, one-tailed Mann–Whitney test with Benjamini-Hochberg correction) is shown. (**D**) Confocal images of pupal eyes (scale bar = 5 μm) stained with anti-DLG illustrate the rescue of ommatidial organization defects due to knockdown of *Cbp20* or *dlg1* upon overexpression of *Diap1*. Yellow circles in DLG images indicate cone cell defects, white circles indicate bristle cell defects, yellow arrows indicate rotation defects, and yellow arrowheads indicate secondary cell defects. We note that pupal eye images were taken at a higher intensity for lines with knockdown of *dlg1* to account for reduced expression of DLG (see Methods), as these images were only for visualization of cell boundaries in the pupal eye and not for any quantitative analysis. (**E**) Larval eye discs (scale bar = 30 μm) stained with anti-dcp1 show rescue of apoptosis phenotypes observed in flies with *Cbp20* and *dlg1* knockdown upon *Diap1* overexpression as well as enhanced (Enh.) phenotypes upon *Dronc* overexpression. (**F**) Boxplot of dcp1-positive cells in the larval eye discs of flies with knockdown of *Cbp20* or *dlg1* and *Diap1* or *Dronc* overexpression (n = 9–18, *p < 0.05, two-tailed Mann–Whitney test with Benjamini-Hochberg correction). (**G**) Representative confocal images of larval eye discs stained with anti-chaoptin (scale bar = 30 μm) illustrate the suppression (Supp.) of axon targeting defects (white arrows) observed in flies due to knockdown of *Cbp20* or *dlg1* with overexpression of *Diap1*. Note that n = 8–18 larval eye disc preparations were assessed for each interaction cross tested. All boxplots indicate median (center line), 25th and 75th percentiles (bounds of box), and minimum and maximum (whiskers), with red dotted lines representing the control median. A list of full genotypes for fly crosses used in these experiments is provided in S2 File.

of *X. laevis* homologs for each 3q29 gene was accomplished using antisense morpholino oligonucleotides (MOs) targeted to early splice sites of each homolog (Fig 1). *X. laevis* embryos were injected at either the two- or four-cell stage with various concentrations of MO for each

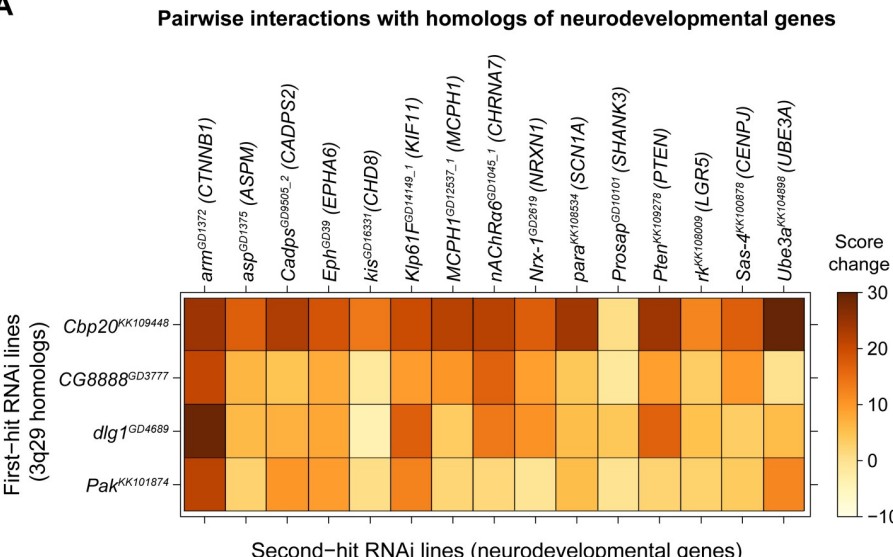

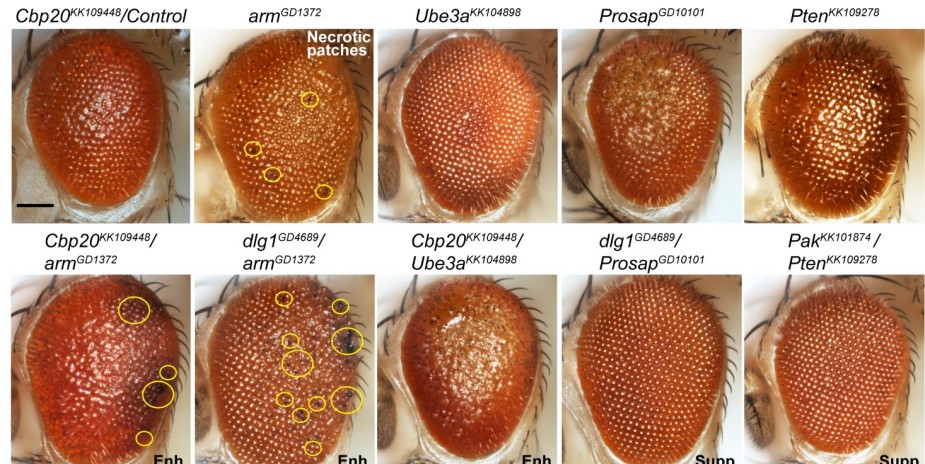

**Fig 7. Pairwise interactions between fly homologs of 3q29 genes and other neurodevelopmental genes.** (**A**) Heatmap showing the average changes in phenotypic scores for the *GMR-GAL4* pairwise RNAi knockdown of fly homologs for 3q29 genes and other neurodevelopmental genes (along with their human counterparts) in the adult eye, compared with recombined lines for individual homologs of 3q29 genes crossed with controls. (**B**) Representative brightfield adult eye images of flies with pairwise knockdown of fly homologs for 3q29 genes and known neurodevelopmental genes (scale bar = 100 μm) show enhancement (Enh.) or suppression (Supp.) of rough eye phenotypes and necrotic patches compared with flies with knockdown of individual homologs of neurodevelopmental genes. A list of full genotypes for fly crosses used in these experiments is provided in **S2 File**.

homolog or a standard control, and knockdown of each homolog was validated using qPCR (**S12 Fig**). As knockdown of *Cbp20*, *Fsn*, and *Pak* each resulted in neuronal defects in *Drosophila*, we first examined the effects of knockdown of these homologs on *X. laevis* brain development at stage 47. To test this, we knocked down each gene in half of the embryo at the two-cell stage, and left the other half uninjected to create a side-by-side comparison of brain morphology (**Fig 8A**). We performed whole-mount immunostaining with anti-alpha tubulin and found that reduced expression of *ncbp2*, *fbxo45*, and *pak2* each resulted in smaller forebrain and midbrain size compared with controls (**Fig 8A–8C**). We also found that simultaneous

**Fig 8. Developmental phenotypes observed with knockdown of homologs of 3q29 genes in *X. laevis* models.** (**A**) To study brain morphology upon knockdown of *X. laevis* homologs of genes in the 3q29 region, one cell in a two-cell embryo was injected with single or multiple MOs for homologs of 3q29 genes while the other cell remained uninjected. Representative images of stage 47 *X. laevis* tadpoles (scale bar = 500 μm) with MO knockdown of *ncbp2*, *fxbo45* and *pak2* show morphological defects and decreased size, including decreased forebrain (highlighted in red on the control image) and midbrain (highlighted in yellow) area, compared with control tadpoles. Pairwise knockdown of *fbxo45* and *ncbp2* enhanced these phenotypes, which were also rescued with overexpression of *xiap*. (**B**) Box plot of forebrain area in *X. laevis* models with knockdown of homologs of 3q29 genes, normalized to controls (n = 30–63, *p < 0.05, two-tailed Welch's T-test with Benjamini-Hochberg correction). Red box indicates rescue of decreased *ncbp2* forebrain area with overexpression of the apoptosis inhibitor *xiap*. (**C**) Box plot of midbrain area in *X. laevis* models with knockdown of homologs of 3q29 genes, normalized to controls (n = 30–63, *p < 0.05, two-tailed Welch's T-test with Benjamini-Hochberg correction). Red box indicates rescue of decreased *ncbp2* midbrain area with overexpression of the apoptosis inhibitor *xiap*. (**D**) Western blot analysis of *X. laevis* whole embryos show increased intensity of cleaved caspase-3 bands at 19kD and 17kD with knockdown of homologs of 3q29 genes, including enhanced caspase-3 levels with knockdown of multiple homologs of 3q29 genes and rescued levels with *xiap* overexpression. β-actin was used as a loading control on the same blot. Representative western blot images shown are cropped; the full blots for both replicates are provided in **S12 Fig**. (**E**) Quantification of western blot band intensity for caspase-3 levels, normalized to the loading control. Red box indicates rescue of increased caspase-3 levels with overexpression of the apoptosis inhibitor *xiap*. All boxplots indicate median (center line), 25th and 75th percentiles (bounds of box), and minimum and maximum (whiskers), with red dotted lines representing the control median. The data shown for the brain area experiments represent pooled results of three experimental batches, and were normalized to the respective controls from each batch. *X. laevis* embryo diagrams were produced by Nieuwkoop and Farber [117] and adapted from Xenbase [120].

knockdown of *ncbp2* with *fbxo45* caused a significant decrease in forebrain size and a trend towards decreased midbrain size (p = 0.093) compared with *ncbp2* knockdown (**Fig 8A–8C**). Knockdown of *pak2* with *ncbp2* showed a similar trend towards decreased forebrain size (p = 0.051). Interestingly, the reduced brain volumes we observed with knockdown of

homologs of 3q29 genes in *X. laevis* recapitulate the reduced brain volume observed in 3q29 deletion mice [14,15], suggesting that multiple genes in the 3q29 region contribute to this phenotype. We further examined the effect of knocking down homologs of 3q29 genes on *X. laevis* eye development at stage 42, and found that knockdown of these homologs caused irregular shapes and decreased size compared with controls (**S13 Fig**). The reductions in eye size were rescued to control levels when mRNA was co-injected along with MO for each homolog (**S13 Fig**). Together, these data show that individual and pairwise knockdown of homologs of 3q29 genes in *X. laevis* leads to abnormal brain and eye morphology, confirming the conserved role of these genes during vertebrate development.

To determine if the knockdown of homologs of 3q29 genes also disrupted apoptotic processes in *X. laevis*, we tested whether overexpression of the X-linked inhibitor of apoptosis gene (*xiap*) could rescue the observed developmental defects. We found that overexpression of *xiap* rescued the midbrain and forebrain size deficits observed with *ncbp2* knockdown to control levels (**Fig 8A–8C**). Similarly, we found that the decreased eye sizes and morphological defects observed with knockdown of *ncbp2* were rescued with *xiap* overexpression (**S13 Fig**). To further validate these findings, we performed a western blot following knockdown of *fbxo45* and *ncbp2* using anti-cleaved caspase-3 (Asp175) as a marker for apoptosis (**Fig 8D, S12 Fig**). We found that reduction of *fbxo45* and *ncbp2* expression each led to an increase in cleaved caspase-3 levels compared with controls, which were restored to control levels with concomitant overexpression of *xiap* (**Fig 8E**). Caspase-3 levels were also enhanced when *fbxo45* and *ncbp2* were knocked down together (**Fig 8E**), suggesting that these two homologs interact with each other and contribute towards developmental phenotypes through increased apoptosis. Overall, these results suggest involvement of apoptotic processes towards the developmental phenotypes observed with knockdown of homologs of 3q29 genes in a vertebrate model (**Table 1**).

## Discussion

Using complementary *Drosophila* and *X. laevis* models, we interrogated developmental effects, cellular mechanisms, and genetic interactions of individual homologs of genes within the 3q29 region. Our major findings were recapitulated across both model systems (**Table 1**) and could also potentially account for the developmental phenotypes reported in mouse models of the entire deletion. Several themes have emerged from our study that exemplify the genetic and mechanistic complexity of the 3q29 deletion region.

*First*, our analysis of developmental phenotypes with knockdown of homologs for individual 3q29 genes showed that a single gene within the region may not be solely responsible for the effects of the deletion. In fact, we found that knockdown of 12 out of 14 fly homologs showed developmental defects in *Drosophila*, while every fly homolog showed an enhanced rough eye phenotype when knocked down along with at least one other homolog (**Fig 2**). Although our study is limited to examining conserved cellular phenotypes of homologs of 3q29 genes in *Drosophila* and *X. laevis*, evidence from other model organisms also supports an oligogenic model for the deletion. In fact, knockout mouse models for several 3q29 genes have been reported to exhibit severe developmental phenotypes, including axonal and synaptic defects in *Fbxo45*<sup>-/-</sup> and embryonic lethality in *Pak2*<sup>-/-</sup> and *Pcyt1a*<sup>-/-</sup> knockout mice [69–71] (**S3 Table**). Notably, although *Dlg1*<sup>+/-</sup> or *Pak2*<sup>+/-</sup> mice showed a range of neuronal phenotypes compared with control mice, they did not recapitulate the major developmental and behavioral features observed in mouse models of the entire deletion [14,15,72], suggesting that the deletion phenotypes are contingent upon haploinsufficiency of multiple genes in the region (**S10 Table**). Furthermore, several 3q29 genes including *PAK2*, *DLG1*, *PCYT1A*, and *UBXN7*

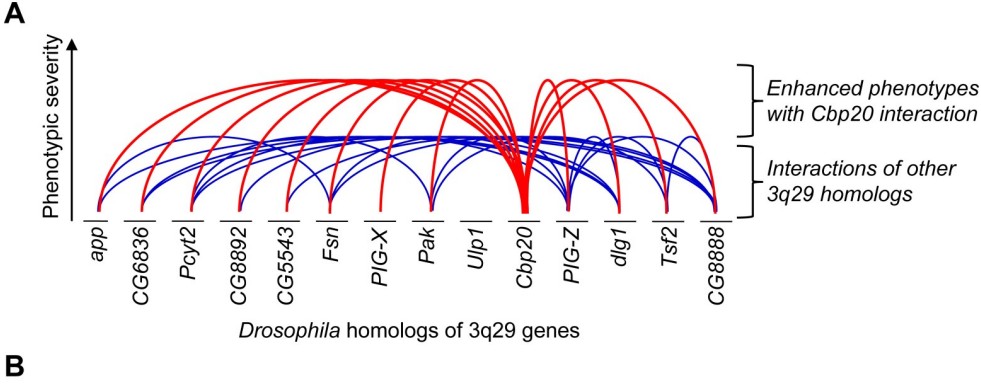

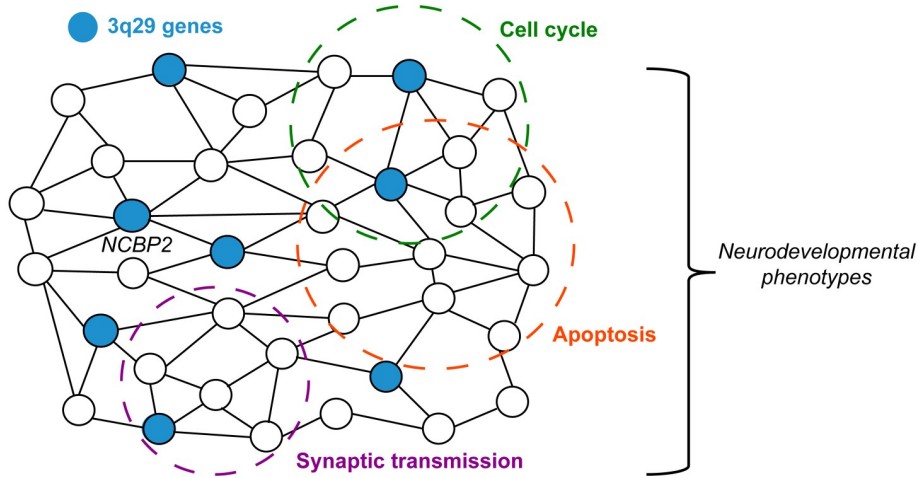

**Fig 9. Interactions between *NCBP2* and other homologs of 3q29 genes contribute to neurodevelopmental defects through conserved cellular pathways. (A)** We identified 44 interactions between pairs of *Drosophila* homologs of 3q29 genes. With the exception of *Ulp1* (*SENP5*), the cellular phenotypes of each homolog were significantly enhanced with simultaneous knockdown of *Cbp20*. While other homologs of 3q29 genes also interact with each other, our data suggest that *Cbp20* is a key modulator of cellular phenotypes within the deletion region. **(B)** Schematic representing the network context of *NCBP2* and other genes in the 3q29 region towards neurodevelopmental phenotypes. We propose that the effects of *NCBP2* disruption propagate through a network of functionally-related genes, including other 3q29 genes (highlighted in blue), leading to a cascade of disruptions in key biological mechanisms, including apoptosis and cell cycle pathways. These pathways jointly contribute towards the observed neurodevelopmental phenotypes.

are under evolutionary constraint in humans, based on gene pathogenicity metrics (**S1 File**). Two genes in the 3q29 region without fly homologs, *CEP19* and *TFRC*, are also under evolutionary constraint in humans, with *TFRC* having been implicated in neural tube defects and embryonic lethality in mouse models [73]. While no common variants associated with neurodevelopmental traits have been observed in the 3q29 region [74], rare variants of varying effects in 9/21 genes have been identified among patients with different developmental disorders [75–77] (**S1 File**). These data, combined with our findings in *Drosophila* and *X. laevis*, implicate multiple genes within the 3q29 region as potential candidates for neurodevelopmental defects.

*Second*, our screening of 161 crosses between pairs of fly homologs of 3q29 genes identified 44 interactions that showed enhanced rough eye phenotypes, suggesting that complex interactions among genes in the 3q29 region could contribute towards developmental defects (**Fig 9A**). While we only tested a subset of all possible interactions among the non-syntenic

homologs of 3q29 genes in *Drosophila*, our results highlight conserved mechanistic relationships between "parts", or the individual genes, towards understanding the effects of the "whole" deletion. For example, knockdown of *Cbp20* enhanced the phenotypes of 11 out of 12 other fly homologs, suggesting that *NCBP2* could be a key modulator of other genes within the region. *NCBP2* encodes a subunit of the nuclear cap-binding complex (CBC), which binds to the 5' end of mRNA and microRNA in the nucleus [78]. Given the role of the CBC in post-transcriptional regulatory mechanisms such as nonsense-mediated decay, alternative splicing and mRNA transport [79,80], it is possible that disruption of this complex could result in changes to a broad set of genes and biological processes. In fact, our analysis of differentially-expressed genes in *Cbp20* knockdown flies showed disruption of synaptic transmission, cellular respiration, and several metabolic pathways. In contrast to other proposed candidate genes in the 3q29 region, *NBCP2* is not predicted to be pathogenic on its own in humans (**S1 File**) and does not have identified deleterious mutations in sequencing studies of neurodevelopmental disease cohorts so far, indicating its potential role as a modifier of the other candidate genes in the region (**Fig 9B**). Our results also complement previous reports of synergistic interactions among fly homologs of 3q29 genes in the nervous system [54], representing another hallmark of an oligogenic model for the deletion. As these genetic interactions may vary across different species, developmental timepoints, and tissues, the role of these interactions should be more deeply explored using mouse and human cell culture models.

*Third*, we identified disruptions to several cellular processes due to both single and pairwise knockdown of homologs in *Drosophila* and *X. laevis* models (**Table 1**). For example, simultaneous knockdown of homologs of *NCBP2* and *FBXO45* in *Drosophila* led to enhanced cellular disorganization (**Fig 5**) and altered expression of cell cycle and apoptosis genes (**S8 Fig**), as well as enhanced morphological defects and increased caspase-3 levels in *X. laevis* (**Fig 8**). We further found that overexpression of the apoptosis inhibitors *Diap1* and *xiap* rescued the cellular and neuronal phenotypes observed with knockdown of homologs of 3q29 genes (**Fig 6**), providing important validations for the potential involvement of apoptosis in the deletion (**Table 1**). We propose that *NCBP2* could modify several cellular and molecular processes that may not be directly related to apoptosis, but could instead lead to a cascade of biological events that ultimately result in apoptosis (**Fig 9B**). Apoptosis mechanisms are well-conserved between *Drosophila*, *X. laevis*, and humans, with key genes such as *XIAP* (*Diap1*), *CASP2* (*Dronc*), *CASP3* (*DrICE*), and *CASP7* (*Dcp-1*) sharing the same roles in programmed cell death across the three organisms [81–83]. In fact, we found that fly homologs of human genes annotated for apoptosis function in the Gene Ontology database were also enriched for apoptosis function (n = 1,063 fly homologs from 1,789 human apoptosis genes; p = $5.30 \times 10^{-13}$, Fisher's Exact test with Benjamini-Hochberg correction). Although we focused on testing apoptosis phenotypes with knockdown of homologs of 3q29 genes, we note that apoptosis is potentially one of many cellular pathways disrupted by the 3q29 deletion (**Fig 9B**). In fact, our data implicated knockdown of several homologs of 3q29 genes, including *dlg1* and *CG8888* (*BDH1*), towards abnormal cell proliferation during development. Furthermore, several 3q29 genes have been previously associated with apoptosis or cell cycle regulation functions (**S1 File**). For example, *DLG1* is a tumor suppressor gene whose knockdown in *Drosophila* leads to neoplasms in the developing brain and eye disc [84,85], while *PAK2* is a key downstream mediator of the ERK signaling pathway for neuronal extension and is activated by caspases during apoptosis [70,86,87]. Our results recapitulate the role of *DLG1* towards cell cycle regulation, and also implicate *NCBP2* and its interactions towards multiple cellular and developmental phenotypes.

More broadly, genes involved with apoptosis and cell proliferation have been implicated in several neurodevelopmental disorders. For example, we previously observed disrupted cell

proliferation upon knockdown of *Drosophila* homologs of genes in the 16p11.2 deletion region, as well as an enrichment of cell cycle function among connector genes between pairs of 16p11.2 genes in a human brain-specific network [21]. Furthermore, abnormal apoptosis in the early developing brain has been suggested as a possible mechanism for the decreased number of neurons observed in individuals with autism and schizophrenia [62,88,89]. For example, increased apoptosis was observed in both postmortem brain tissue from autism patients [90] and primary fibroblasts from schizophrenia patients [91,92]. We found further support for the role of apoptosis in these disorders by identifying significant enrichments for genes associated with apoptotic processes among candidate genes for autism (empirical $p < 1.00 \times 10^{-5}$) [77], intellectual disability ($p < 1.00 \times 10^{-5}$) [93], and schizophrenia ($p = 0.014$) [76] (**S11 Table**). In fact, out of the 525 neurodevelopmental genes involved in apoptosis, 20 genes were present within pathogenic CNV regions [94], including *CORO1A*, *MAPK3* and *TAOK2* in the 16p11.2 region as well as *TBX1*, the causative gene for heart defects in DiGeorge/velocardiofacial syndrome [95] (**S4 File**). In addition to neuropsychiatric disorders, apoptosis has also been implicated in syndromic forms of microcephaly in humans [96] as well as decreased brain size in animal models of microcephaly genes [97,98]. For example, a mouse model of the Nijmegen breakage syndrome gene *NBN* exhibited increased neuronal apoptosis, leading to microcephaly and decreased body mass [99]. Overall, these findings highlight the importance of cell cycle-related processes, particularly apoptosis and proliferation, towards modulating neuronal phenotypes that could be responsible for developmental disorders.

In this study, the use of *Drosophila* and *X. laevis* models, both of which are amenable to high-throughput screening of developmental phenotypes, allowed us to systematically examine the conserved cellular and mechanistic roles of homologs of 3q29 genes and their interactions. Follow-up studies in more evolutionarily advanced systems, such as mouse or human cell lines, will be useful to overcome limitations of *Drosophila* and *X. laevis* models, including testing the neurodevelopmental phenotypes and interactions of 3q29 genes without fly homologs. Collectively, these results emphasize the utility of quantitative functional assays for identifying conserved pathways associated with neurodevelopmental disorders, which will hopefully allow for future discoveries of treatments for these disorders.

## Materials and methods

### Ethics statement

All *X. laevis* experiments were approved by the Boston College Institutional Animal Care and Use Committee (Protocol #2016–012) and were performed according to national regulatory standards.

### Fly stocks and genetics

Using reciprocal BLAST searches and orthology predictions from the DRSC Integrative Ortholog Prediction Tool (*DIOPT*) v.7.1 [100], we identified 15 fly homologs for the 21 human genes within the chromosome 3q29 region (**S1 Table**). No fly homologs were present for six genes, including *LRRC33*, *CEP19*, *RNF168*, *SMCO1*, *TFRC*, and *TM4SF19*. We used a similar strategy to identify homologs for other neurodevelopmental genes tested for interactions in this study. Gene Ontology-Slim (GO-Slim) terms for each human gene and fly homolog were obtained from PantherDB [101] and are provided in **S1 File**. RNAi lines for fly homologs were obtained from the Vienna *Drosophila* Resource Centre [102] (VDRC), including both KK and GD lines, and the Bloomington *Drosophila* Stock Center (BDSC) (NIH P40OD018537). A list of fly RNAi lines used in this study is provided in **S2 File**. Fly RNAi lines for homologs of 3q29 genes were tested for gene knockdown using quantitative PCR (**S1**

Table). As the available KK line for CG5359 (*TCTEX1D2*) showed a wing phenotype consistent with *tiptop* overexpression due to RNAi insertion at the 5'UTR of the gene [103], which we confirmed using qPCR analysis (**S5 File**), we excluded this gene from our experiments. Microarray data and modENCODE Anatomy RNA-Seq from FlyBase [104,105] showed that all of the 14 tested homologs were expressed in the fly central nervous system and eye tissues (**S1 Table**).

All fly stocks and crosses were cultured on conventional cornmeal-sucrose-dextrose-yeast medium at 25˚C, unless otherwise indicated. RNAi lines were crossed with a series of *GAL4* driver lines to achieve tissue-specific knockdown of genes, including $w^{1118}$;*da-GAL4* (Scott Selleck, Penn State) for ubiquitous, $w^{1118}$;*dCad-GFP,GMR-GAL4/CyO* (Zhi-Chun Lai, Penn State) and $w^{1118}$;*GMR-GAL4;UAS-Dicer2* (Claire Thomas, Penn State) for eye-specific, $w^{1118}$, $bx^{MS1096}$-*GAL4;;UAS-Dicer2* (Zhi-Chun Lai, Penn State) for wing-specific, and $w^{1118}$,*Elav-GAL4* (Mike Groteweil, VCU) and $w^{1118}$,*Elav-GAL4;;UAS-Dicer2* (Scott Selleck, Penn State) for pan-neuronal knockdown of gene expression. A list of full genotypes for all crosses tested in this study is provided in **S2 File**. To perform interaction studies, we generated recombined stock lines of *GMR-GAL4* with reduced expression of nine select homologs of 3q29 genes (**S2 File**). Females from these stocks with constitutively reduced gene expression for each of these genes were crossed with RNAi lines of other homologs to achieve simultaneous knockdown of two genes (**Fig 1**). We previously demonstrated that these two-hit crosses had adequate *GAL4* to bind to two independent *UAS-RNAi* constructs [21]. All unique biological materials described in the manuscript, such as recombined fly stocks, are readily available from the authors upon request.

## Quantitative polymerase chain reaction for *Drosophila* RNAi knockdowns

Levels of gene expression knockdown were confirmed using quantitative reverse-transcriptase PCR (qPCR) on RNA isolated from pooled groups of 35 fly heads per line tested (**S2 Table**). Briefly, RNAi lines were crossed with *Elav-GAL4* (to test RNAi line efficacy) or *Elav-GAL4;; UAS-Dicer2* (to test for *tiptop* overexpression) at 25˚C to achieve pan-neuronal knockdown of the fly homolog. Adult fly heads at day 3 were separated by vortexing, and total RNA was isolated using TRIzol (Invitrogen, Carlsbad, CA, USA). cDNA was prepared using the qScript cDNA synthesis kit (Quantabio, Beverly, MA, USA). Quantitative PCR was performed using an Applied Biosystems Fast 7500 system with SYBR Green PCR master mix (Quantabio) to estimate the level of gene expression. All experiments were performed using three biological replicates of 35 fly heads each. Primers were designed using NCBI Primer-BLAST [106], with primer pairs separated by an intron in the corresponding genomic DNA. A list of primers used in the experiments is provided in **S2 Table**. The delta-delta Ct value method was used to obtain the relative expression of fly homologs in the RNAi lines compared with *Elav-GAL4* controls [107].

## Climbing assay

We set up fly crosses at 25˚C with *Elav-GAL4* to obtain pan-neuronal knockdown for select homologs of 3q29 genes. For each RNAi line tested, groups of ten female flies were first allowed to adjust at room temperature for 30 minutes and then transferred to a climbing apparatus, made by joining two vials, and allowed to adjust for 5 minutes. The flies were tapped down to the bottom, and the number of flies climbing past the 8 cm mark measured from the bottom of the apparatus in 10 seconds was then counted (**S1 Video, S2 Video**). This assay was repeated nine additional times for each group, with a one-minute rest between trials. The sets of 10 trials for each group were repeated daily for ten days, capturing data for 100 replicates

from day 1 until day 10, starting the experiments with 1-2-day old flies. All experiments were performed during the same time of the day for consistency of results.

### Imaging of adult fly eyes and wings

We crossed RNAi lines with *GMR-GAL4* and reared at 29˚C for eye-specific knockdown and *bx^{MS1096}-GAL4* at 25˚C for wing-specific knockdown. For eye imaging, adult 2-3-day old female progenies from the crosses were collected, immobilized by freezing at -80˚C, mounted on Blu-tac (Bostik Inc, Wauwatosa, WI, USA), and imaged with an Olympus BX53 compound microscope with LMPLan N 20X air objective using a DP73 c-mount camera at 0.5X magnification and a z-step size of 12.1μm. (Olympus Corporation, Tokyo, Japan). We used CellSens Dimension software (Olympus Corporation, Tokyo, Japan) to capture the images, and stacked the image slices using Zerene Stacker (Zerene Systems LLC, Richland, WA, USA). All eye images presented in this study are maximum projections of 20 consecutive optical z-sections. Adult wings were plucked from 2–5 day old female flies, mounted on a glass slide, covered with a coverslip and sealed with clear nail polish. The wings were imaged using a Zeiss Discovery V20 stereoscope (Zeiss, Thornwood, NY, USA) with ProgRes Speed XT Core 3 camera (Jenoptik AG, Jena, Germany) using a 40X objective, and wing images were captured with ProgRes CapturePro v.2.8.8 software.

### Quantitative phenotyping of fly eyes using *Flynotyper*

We used a computational method called *Flynotyper* (http://flynotyper.sourceforge.net) to measure the degree of roughness of the adult eyes with knockdown of individual or pairs of homologs [53]. The software uses an algorithm to detect the center of each ommatidium, and calculates a phenotypic score based on the number of ommatidia detected, the lengths of six local vectors with direction pointing from each ommatidium to the neighboring ommatidia, and the angle between these six local vectors (S2 Fig). Eye areas, ommatidial diameter, and areas of necrotic patches, which may not be reflected in the *Flynotyper* scores, were measured using ImageJ [108]. Significant pairwise interactions were reported as "validated" when multiple RNAi or mutant lines, if available, showed the same phenotype (S7 Table, S9 Table).

### Immunohistochemistry of eye and wing discs

Third instar larval and 44-hour-old pupal eye discs, reared at 29˚C, and third instar larval wing discs, reared at 25˚C, were dissected in 1X phosphate-buffered saline (PBS) and fixed in 4% paraformaldehyde for 20 minutes. The eye and wing discs were then washed thrice in PBT (PBS with 0.1% Triton-X) for 10 minutes each, treated with blocking solution (PBS with 1% normal goat serum (NGS) for eye discs, or 1% bovine serum albumin (BSA) for wing discs) for 30 minutes, and then incubated overnight with primary antibodies at 4˚C. Rabbit anti-cleaved *Drosophila* dcp1 (Asp216) (1:100; 9578S, Cell Signaling Technology, Danvers, MA, USA), a marker for cells undergoing apoptosis, and Mouse anti-phospho-Histone H3 (S10) antibody (1:100; 9706L, Cell Signaling Technology), a mitotic marker for measuring proliferating cells, were used to assay cell proliferation and apoptosis defects in larval eye and wing discs. Mouse anti-DLG (1:200; 4F3, DSHB, Iowa City, Iowa, USA), a septate junction marker, and Rhodamine Phalloidin (1:200; R415, Invitrogen Molecular Probes, Carlsbad, CA, USA), an F-actin marker, were used to visualize and count ommatidial cells and photoreceptor cells in pupal eyes. Mouse anti-chaoptin (1:200; 24B10, DSHB) was used to visualize retinal axon projections. Preparations were then washed thrice with PBT for 10 minutes, and incubated for two hours with fluorophore-conjugated secondary antibodies (Alexa fluor 568 goat anti-mouse (1:200) (A11031), Alexa fluor 488 goat anti-mouse (1:200) (A11029), Alexa fluor 647 goat anti-

rabbit (1:200) (A21245), and Alexa fluor 647 goat anti-mouse (1:200) (A21236), Invitrogen Molecular Probes, Carlsbad, CA, USA)) with gentle shaking. Preparations were washed thrice in PBT for 10 minutes, and the tissues were then mounted in Prolong Gold antifade mounting media with DAPI (P36930, Thermo Fisher Scientific, Waltham, MA, USA) or Vectashield hard set mounting media with DAPI (H-1500, Vector Laboratories, Burlingame, CA, USA) for imaging.

### Bromouridine staining

Third instar larval eye discs were dissected in 1X PBS and immediately transferred to Schneider's Insect Media (Sigma-Aldrich, St. Louis, MO). The tissues were then incubated in 10 μM BrdU (Sigma-Aldrich) at 25˚C for one hour with constant agitation to allow for incorporation of BrdU into DNA of replicating cells during the S-phase of cell cycle. The samples were washed thrice with PBS for five minutes each and fixed in 4% paraformaldehyde for 20 minutes. To denature DNA, the tissues were acid-treated in 2N HCl for 20 minutes, neutralized in 100 mM Borax solution for 2 minutes, washed thrice in 10X PBT (PBS with 0.1% Tween-20) for 10 minutes, and treated with blocking solution (PBS, 0.2% Triton X-100, 5% NGS) for one hour. The tissues were then incubated in mouse anti-BrdU (1:200; G3G4, DSHB, Iowa City, Iowa, USA) and diluted in blocking solution overnight at 4˚C. The next day, the tissues were washed thrice in PBT for 20 minutes each and incubated in Alexa fluor 568 goat anti-mouse (1:200, Invitrogen Molecular Probes, Carlsbad, CA, USA) for two hours with constant agitation. Finally, the samples were mounted in Prolong Gold antifade reagent with DAPI (Thermo Fisher Scientific, Waltham, MA, USA) for imaging.

### Terminal deoxynucleotidyl transferase (TUNEL) Assay

The levels of cell death in the developing eye were evaluated by staining using the *In Situ* Cell Death Detection Kit, TMR Red (Roche, Basel, Switzerland). The third instar larval eye discs were dissected in 1X PBS and fixed in 4% paraformaldehyde for 20 minutes at room temperature, followed by three 10-minute washes with PBS. The dissected tissues were permeabilized by treating with 20 μg/ml proteinase K (Sigma-Aldrich, St. Louis, MO, USA) for two minutes, washed thrice in PBT (PBS with 0.1% Triton-X) for 5 minutes each, fixed in 4% paraformaldehyde for 15 minutes, and washed thrice again in PBT for 10 minutes each. The tissues were then incubated overnight with TUNEL (terminal deoxynucleotidyl transferase dUTP nick end labeling) reaction mixture at 4˚C per the manufacturer's instructions, and washed five times in PBT for 15 minutes each. Finally, tissues were mounted in Prolong Gold antifade containing DAPI (Thermo Fisher Scientific, Waltham, MA, USA) for imaging.

### Confocal imaging and analysis

Confocal images of larval and pupal eye and wing discs were captured using an Olympus Fluoview FV1000 laser scanning confocal microscope (Olympus America, Lake Success, NY). Maximum projections of all optical sections were generated for display. As DLG staining was only used to visualize cell boundaries in the pupal eye and not for any expression or quantitative analysis, we increased the laser intensity from 400-490V in control flies to 530-570V in flies with knockdown of *dlg1* to account for decreased DLG expression. Acquisition and processing of images was performed using the Fluoview FV10-ASW 2.1 software (Olympus Corporation, Tokyo, Japan), and the z-stacks of images were merged using ImageJ [108]. The number of pH3, BrdU, TUNEL, and dcp1-positive cells from larval eye discs were counted using two ImageJ plugins, AnalyzeParticles and Image-based Tool for Counting Nuclei (ITCN). As we found a strong correlation (Pearson correlation, $r = 0.736$, $p<2.2x10^{-16}$) between the two methods

(**S2 Fig**), all cell counts displayed for eye data were derived from ITCN analysis. Proliferating cells in larval wing discs stained with pH3 were counted using AnalyzeParticles, and apoptotic cells in wing discs stained with dcp1 were analyzed using manual counting. Images stained with anti-chaoptin were manually scored as having either "mild" (minor axon disorganization compared with control), "moderate" (partial loss of axon projection. i.e. loss of R7-R8 projection into the medulla), or "severe" (loss of projections for most axons at the lamina) axon targeting defects.

## Differential expression analysis of transcriptome data

We performed RNA sequencing (RNA-Seq) of samples isolated from three biological replicates of 35 fly heads each for individual (*Cbp20*, *dlg1*, *Fsn*, *Pak*) and pairwise (*Cbp20/dlg1*, *Cbp20/Fsn*) *Elav-GAL4* mediated knockdowns of homologs of 3q29 genes. We compared gene expression levels of each cross to VDRC control flies carrying the same genetic background (GD or KK control lines crossed with *Elav-GAL4*). We prepared cDNA libraries for the three biological replicates per genotype using TruSeq Stranded mRNA LT Sample Prep Kit (Illumina, San Diego, CA), and performed single-end sequencing using Illumina HiSeq 2000 at the Penn State Genomics Core Facility to obtain 100 bp reads at an average coverage of 36.0 million aligned reads/sample. We used Trimmomatic v.0.36 [109] for quality control assessment, TopHat2 v.2.1.1 [110] to align the raw sequencing data to the reference fly genome and transcriptome (build 6.08), and HTSeq-Count v.0.6.1 [111] to calculate raw read counts for each gene. edgeR v.3.20.1 [112] (generalized linear model option) was used to perform differential expression analysis, and genes with $\log_2$-fold changes $>1$ or $<-1$ and false-discovery rates $<0.05$ (Benjamini-Hochberg correction) were considered to be differentially expressed (**S3 File**). Human homologs of differentially-expressed fly genes (top matches for each fly gene, excluding matches with "low" rank) were identified using DIOPT [100]. Enrichment analysis of Panther GO-Slim Biological Process terms among the differentially-expressed fly genes and their human homologs was performed using the PantherDB Gene List Analysis tool [101]. Enrichments for genes preferentially expressed in the developing brain were calculated using the Cell-type Specific Expression Analysis tool [113] based on expression data from the BrainSpan Atlas [114].

## *X. laevis* embryos

Eggs collected from female *X. laevis* frogs were fertilized *in vitro*, dejellied, and cultured following standard methods [115,116]. Embryos were staged according to Nieuwkoop and Faber [117].

## Morpholino and RNA constructs

Morpholinos (MOs) were targeted to early splice sites of *X. laevis* *ncbp2*, *fbxo45*, *pak2*, or standard control MO, purchased from Gene Tools LLC (Philomath, OR, USA). MO sequences are listed in **S12 Table**. For knockdown experiments, all MOs were injected at either the 2-cell or 4-cell stage, with embryos receiving injections two or four times total in 0.1X MMR media containing 5% Ficoll. Control and *fbxo45* MOs were injected at 10ng/embryo, *ncbp2* and control MOs were injected at 20ng/embryo, and *pak2* and control MOs were injected at 50ng/embryo. For rescue experiments (**S13 Fig**), the same amounts of MOs used in the KD experiments were injected along with gene-specific mRNA tagged with GFP (800pg/embryo for *xiap*-GFP; 1000pg/embryo for *ncbp2*-GFP and *fbxo45*-GFP, and 300pg/embryo for *pak2*-GFP) in the same injection solution. Capped mRNAs were transcribed *in vitro* using SP6 or T7 mMessage mMachine Kit (Thermo Fisher Scientific, Waltham, MA, USA). RNA was purified

with LiCl precipitation. *X. laevis ncbp2*, *fbxo45*, *pak2*, and *xiap* ORFs obtained from the European *Xenopus* Resource Center (EXRC, Portsmouth, UK) were gateway-cloned into pCSf107mT-GATEWAY-3'GFP destination vectors. Constructs used included *ncbp2*-GFP, *fbxo45*-GFP, *pak2*-GFP, *xiap*-GFP, and GFP in pCS2+. Embryos either at the 2-cell or 4-cell stage received four injections in 0.1X MMR containing 5% Ficoll with the following total mRNA amount per embryo: 300pg of GFP, 800pg of *xiap*-GFP, 1000pg of *ncbp2*-GFP, 1000pg of *fbxo45*-GFP, and 300pg of *pak2*-GFP.

### qPCR for *X. laevis* morpholino knockdown

Morpholino validation and knockdown was assessed using qPCR. Total RNA was extracted using TRIzol reagent (Life Technologies, Grand Island, NY, USA), followed by chloroform extraction and ethanol precipitation from 2-day old embryos injected with increasing concentrations of MO targeted to each homolog of the tested 3q29 gene. cDNA synthesis was performed with SuperScript II Reverse Transcriptase (Life Technologies, Grand Island, NY, USA) and random hexamers. PCR primers are listed in **S13 Table**. qPCR was performed in triplicate (**S12 Fig**), with band intensities quantified by densitometry in ImageJ and normalized to the uninjected control mean relative to *ODC1*, which was used as a housekeeping control.

### Brain and eye morphology assays

In brain morphology experiments, all embryos received two injections at the 2-cell stage in 0.1X MMR containing 5% Ficoll. One cell was left uninjected and the other cell was injected with either control MO or MO targeted to the tested 3q29 gene, along with 300pg of GFP mRNA in the same injection solution. Stage 47 tadpoles were fixed in 4% PFA diluted in PBS for one hour, rinsed in PBS and gutted to reduce autofluorescence. Embryos were incubated in 3% bovine serum albumin and 1% Triton-X 100 in PBS for two hours, and then incubated in anti-acetylated tubulin primary antibody (1:500, monoclonal, clone 6-11B-1, AB24610, Abcam, Cambridge, UK) and goat anti-mouse Alexa fluor 488 conjugate secondary antibody (1:1000, polyclonal, A11029, Invitrogen Life Technologies, Carlsbad, CA). Embryos were then rinsed in 1% PBS-Tween and imaged in PBS. Skin dorsal to the brain was removed if the brain was not clearly visible due to pigment. For eye phenotype experiments, all embryos received four injections at the 2-cell or 4-cell stage in 0.1X MMR containing 5% Ficoll with either the control MO or MOs targeted to each 3q29 gene. Stage 42 tadpoles were fixed in 4% PFA diluted in PBS. Tadpoles were washed three times in 1% PBS-Tween for one hour at room temperature before imaging.

### *X. laevis* image acquisition and analysis

Lateral view images of stage 42 tadpoles for eye experiments and dorsal view images of state 47 tadpoles for brain experiments were each collected on a SteREO Discovery.V8 microscope using a Zeiss 5X objective and Axiocam 512 color camera (Zeiss, Thornwood, NY, USA). Areas of the left and right eye, forebrain, and midbrain were determined from raw images using the polygon area function in ImageJ. Eye size was quantified by taking the average area of both the left and right eye, while forebrain and midbrain area were quantified by taking the ratio between the injected and uninjected sides for each sample.

### Western blot for apoptosis

Two replicate western blot experiments were performed to test for apoptosis markers in *X. laevis* with 3q29 gene knockdown (**S12 Fig**). Embryos at stages 20–22 were lysed in buffer (50mM Tris pH 7.5, 1% NP40, 150mM NaCl, 1mM PMSF, 0.5 mM EDTA) supplemented

with cOmplete Mini EDTA-free Protease Inhibitor Cocktail (Sigma-Aldrich, Basel, Switzerland). Blotting was carried out using rabbit polyclonal antibody to cleaved caspase-3 (1:500, 9661S, Cell Signaling Technology, Danvers, MA, USA), with mouse anti-beta actin (1:2500, AB8224, Abcam, Cambridge, UK) as a loading control on a Mini-PROTEAN TGX precast 4–15% gradient gel (Bio-Rad, Hercules, CA, USA). Chemiluminescence detection was performed using Amersham ECL western blot reagent (GE Healthcare Bio-Sciences, Pittsburgh, PA, USA). Band intensities were quantified by densitometry in ImageJ and normalized to the control mean relative to beta-actin. Due to the low number of replicates, we did not perform any statistical tests on data derived from these experiments.

### Human brain-specific network analysis of 3q29 gene interactions

We used a human brain-specific gene interaction network that was previously built using a Bayesian classifier trained on gene co-expression datasets [56,57]. We extracted interactions between pairs of genes with predicted weights >2.0 (containing the top 0.5% most likely interactions) and measured the length of the shortest paths connecting pairs of 3q29 genes within the network, excluding genes not present in the network from final calculations. As a control, we also measured the connectivity of 500 randomly selected genes with 100 replicates each of 20 other random genes. All network analysis was performed using the NetworkX Python package [118].

### Overlap between neurodevelopmental and apoptosis gene sets

We obtained a set of 1,794 genes annotated with the Gene Ontology term for apoptotic processes (GO:0006915) or children terms from the Gene Ontology Consortium (AmiGO v.2.4.26) [119], and overlapped this gene set with sets of 756 candidate autism genes (SFARI Gene Tiers 1–4) [77], 1,854 candidate intellectual disability genes [93], and 2,546 curated candidate schizophrenia genes [76]. Genes in these three sets that were annotated for apoptosis function are listed in **S4 File**. To determine the statistical significance of these overlaps, we performed 100,000 simulations to identify the number of apoptosis genes among groups of genes randomly selected from the genome, and determined the percentiles for each observed overlap among the simulated overlaps as empirical p-values.

### Statistical analysis

Details of each dataset and the associated statistical tests are provided in **S5 File**. All statistical analyses of functional data were performed using R v.3.4.2 (R Foundation for Statistical Computing, Vienna, Austria). Non-parametric one-tailed and two-tailed Mann-Whitney tests were used to analyze *Drosophila* functional data and human network data, as several datasets were not normally distributed (p<0.05, Shapiro-Wilk tests for normality). Climbing ability and survival data for each fly RNAi line across each experiment day were analyzed using two-way and one-way repeated values ANOVA tests with post-hoc pairwise t-tests. We also used parametric t-tests to analyze *Drosophila* qPCR data and all *X. laevis* data, as these data were either normally distributed (p>0.05, Shapiro-Wilk tests for normality) or had a robust sample size (n>30) for non-normality. All p-values from statistical tests derived from similar sets of experiments (i.e. *Flynotyper* scores for pairwise interactions, dcp1 rescue experiments with *Diap1*) were corrected using Benjamini-Hochberg correction.

### Reproducibility

*Drosophila* eye area and pH3 and TUNEL staining experiments for select individual knock-down lines, as well as climbing ability experiments for a subset of individual and pairwise

knockdown lines, were performed on two independent occasions with similar sample sizes. Data displayed in the main figures were derived from single batches, while data from the repeated experiments are shown in **S14 Fig**. *X. laevis* brain and eye area experiments were performed on three independent occasions, with the data shown in the figures representing pooled results of each of the three experimental batches (normalized to the respective controls from each batch). *X. laevis* qPCR experiments were performed three times and western blot experiments were performed twice, with the blots/gels for each replicate experiment shown in **S12 Fig**. Sample sizes for each experiment were determined by testing all available organisms; no prior power calculations for sample size estimation were performed. No data points or outliers were excluded from the experiments presented in the manuscript.

## Code availability

All source code and datasets for generating genomic data (RNA-Seq, network analysis, and neurodevelopment/apoptosis gene overlap) are available on the Girirajan lab GitHub page at https://github.com/girirajanlab/3q29_project.

## Supporting information

**S1 Fig. Developmental defects in flies with tissue-specific knockdown of individual homologs of 3q29 genes.** (**A**) Images of adult fly wings (scale bar = 500um) show a range of phenotypic defects due to wing-specific $bx^{MS1096}$-*GAL4* RNAi knockdown of fly homologs of 3q29 genes. (**B**) Adult flies with pan-neuronal RNAi knockdown of *dlg1* showed approximately 30% lethality between days 1–4 (one-way repeated measures ANOVA, $p < 1 \times 10^{-4}$, df = 1, F = 54.230), which was not observed in control *Elav-GAL4* or *Cbp20* knockdown flies. Data represented shows mean ± standard deviation of 10 independent groups of 10 flies for each homolog. (**C**) Representative confocal images of larval eye discs stained with anti-chaoptin (scale bar = 30 μm) illustrate defects in axon targeting (highlighted by white arrows) from the retina to the optic lobes of the brain upon eye-specific knockdown of fly homologs of 3q29 genes. Note that n = 8–20 larval eye disc preparations were assessed for each RNAi line tested. A list of full genotypes for fly crosses used in these experiments is provided in **S2 File**. (PDF)

**S2 Fig. Examination of cellular phenotypes in the *Drosophila* eye.** We tested individual and pairwise knockdown of fly homologs of 3q29 genes for cellular phenotypes in the adult, pupal and larval eyes. (**A**) We first used the *Flynotyper* software [53] to quantify the degree of ommatidial disorganization leading to rough eye phenotypes in adult flies, as represented by the distance and angles between adjacent ommatidia (yellow arrows). (**B**) We next stained pupal eyes with anti-DLG to observe changes in the number and arrangement of ommatidial cells, including cone cells (c), bristle cells (b), and primary, secondary and tertiary cells (1,2,3). We also examined the organization of the photoreceptor cells (R1-R7, with R8 not visible) in each ommatidium by staining the pupal eyes with Phalloidin. (**C**) We finally stained larval eye discs with markers for cellular processes, such as pH3 for proliferating cells and dcp1 for apoptosis. As the progression of the morphogenetic furrow (MF) across the larval eye discs leads to proliferation and differentiation of photoreceptor neurons [121], we examined changes in the number of stained cells posterior to the MF. (**D**) Scatter plot of dcp1, pH3, TUNEL, and BrdU-positive cell counts in larval eye discs with knockdown of homologs of 3q29 genes quantified using two ImageJ plugins, AnalyzeParticles and Image-based Tool for Counting Nuclei (ITCN). As the two methods showed a strong correlation with each other (Pearson correlation, n = 285, r = 0.736, $p < 2.2 \times 10^{-16}$), we used ITCN counts to display cell count data in the

manuscript.
(PDF)

**S3 Fig. Phenotypic screening for flies with eye-specific knockdown of individual fly homologs of 3q29 genes.** (**A**) Representative brightfield adult eye images of flies with *GMR-GAL4; UAS-Dicer2* RNAi knockdown of fly homologs of 3q29 genes (scale bar = 100 μm) show a wide range of phenotypic severity. (**B**) Box plot of average ommatidial diameter in flies with *GMR-GAL4* knockdown of select fly homologs of 3q29 genes is shown (n = 15, *p < 0.05, two-tailed Mann–Whitney test with Benjamini-Hochberg correction). (**C**) Box plot of phenotypic scores derived from *Flynotyper* for eye-specific *dCad-GFP,GMR-GAL4* RNAi knockdown of 13 fly homologs of 3q29 genes is shown (n = 5–20, *p < 0.05, one-tailed Mann–Whitney test with Benjamini-Hochberg correction). (**D**) Box plot of phenotypic scores derived from *Flynotyper* for eye-specific *GMR-GAL4;UAS-Dicer2* (left) and *dCad-GFP,GMR-GAL4* (right) RNAi knockdown of nine validation lines for fly homologs of 3q29 genes is shown (n = 5–14, *p < 0.05, one-tailed Mann–Whitney test with Benjamini-Hochberg correction). All boxplots indicate median (center line), 25th and 75th percentiles (bounds of box), and minimum and maximum (whiskers), with red dotted lines representing the control median. A list of full genotypes for fly crosses used in these experiments is provided in **S2 File**.
(PDF)

**S4 Fig. Cellular phenotypes of flies with eye-specific knockdown of individual fly homologs of 3q29 genes.** (**A**) Confocal images of pupal eyes (scale bar = 5 μm) stained with anti-DLG illustrate a range of defects in ommatidial organization upon *GMR-GAL4* RNAi knockdown of fly homologs of 3q29 genes. Yellow circles indicate cone cell defects, white circles indicate bristle cell defects, yellow arrows indicate rotation defects, and yellow arrowheads indicate secondary cell defects. (**B**) Confocal images of pupal eyes (scale bar = 5 μm) stained with Phalloidin illustrate defects in photoreceptor cell count and organization upon knockdown of fly homologs of 3q29 genes. (**C**) Confocal images of larval eye discs (scale bar = 30 μm) stained with anti-pH3 illustrate changes in cell proliferation upon knockdown of select fly homologs of 3q29 genes. (**D**) Larval eye discs (scale bar = 30 μm) stained with BrdU (top) and TUNEL (bottom) illustrate abnormal cell cycle and apoptosis defects, respectively, due to eye-specific knockdown of *Cbp20* and *dlg1*. (**E**) Box plot of BrdU-positive cells in the larval eye discs of flies with knockdown of *dlg1* and *Cbp20* is shown (n = 7–12, *p < 0.05, two-tailed Mann–Whitney test with Benjamini-Hochberg correction). (**F**) Box plot of TUNEL-positive cells in the larval eye discs of flies with knockdown of *dlg1* and *Cbp20* is shown (n = 8, *p < 0.05, two-tailed Mann–Whitney test with Benjamini-Hochberg correction). Results for the TUNEL staining experiments were replicated in an independent experimental batch (**S14 Fig**). All boxplots indicate median (center line), 25th and 75th percentiles (bounds of box), and minimum and maximum (whiskers), with red dotted lines representing the control median. A list of full genotypes for fly crosses used in these experiments is provided in **S2 File**.
(PDF)

**S5 Fig. Cellular phenotypes of flies with wing-specific knockdown of individual fly homologs of 3q29 genes.** (**A**) Larval wing discs (scale bar = 50 μm) stained with pH3 illustrate abnormal cell proliferation due to RNAi knockdown of select fly homologs of 3q29 genes, compared with appropriate VDRC GD and KK *bx^{MS1096}-GAL4* controls. We examined changes in the number of stained cells within the wing pouch of the wing disc (white box), which becomes the adult wing. (**B**) Box plot of pH3-positive cells in the larval wing discs of flies with knockdown of select fly homologs of 3q29 genes is shown (n = 8–15, *p < 0.05, two-tailed Mann–Whitney test with Benjamini-Hochberg correction). (**C**) Larval wing discs (scale

bar = 50 μm) stained with anti-dcp1 show abnormal apoptosis due to knockdown of select fly homologs of 3q29 genes compared with appropriate VDRC GD and KK *bx^MS1096*-GAL4 controls. (**D**) Box plot of dcp1-positive cells in the larval wing discs of flies with knockdown of select fly homologs of 3q29 genes is shown (n = 8–15, *p < 0.05, two-tailed Mann–Whitney test with Benjamini-Hochberg correction). *Cbp20* flies showed severe dcp1 staining across the entire wing pouch and could not be quantified. All boxplots indicate median (center line), 25th and 75th percentiles (bounds of box), and minimum and maximum (whiskers), with red dotted lines representing the control median. A list of full genotypes for fly crosses used in these experiments is provided in **S2 File**.
(PDF)

**S6 Fig. Phenotypic screening for pairwise interactions of homologs of 3q29 genes in the adult fly eye.** (**A**) Heatmap showing average changes in phenotypic scores for pairwise *GMR-GAL4* RNAi knockdown of fly homologs of 3q29 genes in the adult eye, compared with recombined lines for individual homologs of 3q29 genes crossed with controls, is shown. Gray boxes indicate crosses without available data. Crosses with the mutant line *Tsf2^KG01571* are also included along with RNAi lines for other homologs of 3q29 genes, as eye-specific RNAi knockdown of *Tsf2* was lethal. (**B-H**) Box plots of phenotypic scores for pairwise knockdowns of homologs of 3q29 genes compared with recombined lines for individual homologs of 3q29 genes crossed with controls are shown (n = 5–12, *p < 0.05, two-tailed Mann–Whitney test with Benjamini-Hochberg correction). All boxplots indicate median (center line), 25th and 75th percentiles (bounds of box), and minimum and maximum (whiskers), with red dotted lines representing the control median. A list of full genotypes for fly crosses used in these experiments is provided in **S2 File**.
(PDF)

**S7 Fig. Validation lines for pairwise interactions of homologs of 3q29 genes in the adult fly eye.** (**A-F**) Box plots of phenotypic scores for pairwise *GMR-GAL4* RNAi knockdown of select fly homologs of 3q29 genes (*Cbp20*, *CG8888*, *dlg1*, *Fsn*, *Pak*, and *PIG-Z*) with validation RNAi and mutant lines for other homologs of 3q29 genes, compared with recombined lines for individual homologs of 3q29 genes crossed with controls, are shown (n = 4–14, *p < 0.05, two-tailed Mann–Whitney test with Benjamini-Hochberg correction), are shown. These crosses include flies homozygous for *Cbp20* RNAi as well as flies homozygous for *Cbp20* RNAi and heterozygous for *dlg1* RNAi (green arrows). Note that the phenotypic scores derived from *Fly-notyper* may not accurately capture the necrotic patches observed in these crosses. All boxplots indicate median (center line), 25th and 75th percentiles (bounds of box), and minimum and maximum (whiskers), with red dotted lines representing the control median. A list of full genotypes for fly crosses used in these experiments is provided in **S2 File**.
(PDF)

**S8 Fig. Transcriptome analysis of flies with knockdown of select homologs of 3q29 genes.** (**A**) Clusters of Gene Ontology terms enriched among differentially-expressed fly genes (blue) and their corresponding human homologs (red) with individual and pairwise *Elav-GAL4* RNAi knockdown of fly homologs of 3q29 genes (p< 0.05, Fisher's Exact test with Benjamini-Hochberg correction) are shown. Black boxes indicate enrichment of each gene set for clusters of Gene Ontology terms. Full lists of enriched GO terms are provided in **S3 File**. (**B**) Enrichments for shared and unique differentially-expressed fly genes (blue) and their corresponding human homologs (red) with individual knockdown of *Cbp20* and *Fsn*, as well as concomitant knockdown of *Cbp20/Fsn*, are shown. We found 229 genes uniquely dysregulated in flies with pairwise knockdown of *Fsn and Cbp20*, which were enriched for cell cycle function (p = 0.011

for fly gene enrichment and p = $1.12 \times 10^{-8}$ for human homologs, Fisher's Exact test with Benjamini-Hochberg correction). (**C**) Diagram showing human cell cycle and apoptosis genes whose fly homologs are differentially expressed with knockdown of *Cbp20* and *Fsn*, as well as concomitant knockdown of *Cbp20/Fsn*. Red boxes indicate apoptosis genes, green boxes indicate cell cycle genes, and yellow boxes indicate genes associated with both functions. (**D**) Enrichments of human homologs of genes differentially expressed in flies with knockdown of *Cbp20/Fsn* across different brain tissues and developmental timepoints are shown (Specific Expression Analysis). The size of each hexagon represents the number of genes preferentially expressed at each tissue and timepoint, with concentric hexagons representing bins of genes with stronger levels of preferential expression. The shading of each hexagon represents the enrichment of differentially-expressed genes among genes preferentially expressed at each timepoint (p<0.1, Fisher's Exact test with Benjamini-Hochberg correction). A list of full genotypes for fly crosses used in these experiments is provided in **S2 File**.
(PDF)

**S9 Fig. Cellular phenotypes for pairwise knockdowns of homologs of 3q29 genes.** (**A**) Box plot showing the area of necrotic patches in adult fly eyes with pairwise knockdown of homologs of 3q29 genes (n = 5–13, *p < 0.05, one-tailed Mann–Whitney test with Benjamini-Hochberg correction). Flies with knockdown of *Cbp20* and *Fsn*, *dlg1* and *arm* showed enhanced necrotic patches compared with knockdown of *Cbp20*, while homozygous *Cbp20* RNAi and concomitant knockdown of *dlg1* showed increased necrotic patches compared with homozygous *Cbp20* RNAi. Furthermore, flies with knockdown of *dlg1* and *arm* both showed enhanced necrotic patches compared with individual knockdown of *dlg1* or *arm*. (**B**) Confocal images of pupal eyes (scale bar = 5 μm) stained with anti-DLG (top) and Phalloidin (bottom) illustrate enhanced defects in ommatidial and photoreceptor cell organization with concomitant *GMR-GAL4* RNAi knockdown of *Cbp20* and other fly homologs of 3q29 genes compared with *Cbp20* knockdown. (**C**) Larval eye discs (scale bar = 30 μm) stained with TUNEL show increases in apoptosis with pairwise knockdown of *Cbp20* and other fly homologs of 3q29 genes compared with recombined *Cbp20* knockdown crossed with control. (**D**) Box plot of TUNEL-positive cells in the larval eye discs of flies with pairwise knockdown of homologs of 3q29 genes (n = 9–13, *p < 0.05, two-tailed Mann–Whitney test with Benjamini-Hochberg correction). All boxplots indicate median (center line), 25th and 75th percentiles (bounds of box), and minimum and maximum (whiskers), with red dotted lines representing the control median. A list of full genotypes for fly crosses used in these experiments is provided in **S2 File**.
(PDF)

**S10 Fig. Rescue of cellular phenotypes due to knockdown of fly homologs of 3q29 genes with overexpression of *Diap1*.** (**A**) Cellular phenotypes of flies with overexpression of *Diap1* and *Dronc*. Representative brightfield adult eye images (scale bar = 100 μm) and confocal images of larval eye discs (scale bar = 30 μm) stained with anti-dcp1 are shown for flies with *GMR-GAL4* overexpression of *Diap1* and *Dronc*, while confocal images of pupal eyes (scale bar = 5 μm) stained with anti-DLG are also shown for flies with overexpression of *Diap1*. While the overexpression of *Diap1* did not lead to any changes in the pupal or adult eye phenotype, overexpression of *Dronc* resulted in a large increase in apoptosis and depigmentation in the adult eye. (**B**) Box plot of *Flynotyper* distance ommatidial disorderliness (OD) scores for flies with concomitant *GMR-GAL4* RNAi knockdown of *Cbp20* or *dlg1* and overexpression of *Diap1* or *Dronc* is shown (n = 8–9, *p < 0.05, two-tailed Mann–Whitney test with Benjamini-Hochberg correction). (**C**) Box plot of *Flynotyper* angle OD scores for flies with knockdown of *Cbp20* or *dlg1* and overexpression of *Diap1* or *Dronc* is shown (n = 8–9, *p < 0.05, two-tailed Mann–Whitney test with Benjamini-Hochberg correction). The distance and angle OD scores,

component subscores derived from *Flynotyper* [53], mirror the trends observed in the overall phenotypic scores (**Fig 6B**). (**D**) Box plot of adult eye area in flies with knockdown of *Cbp20* or *dlg1* and overexpression of *Diap1* or *Dronc* is shown (n = 8–9, *p < 0.05, two-tailed Mann–Whitney test with Benjamini-Hochberg correction). (**E**) Confocal images of pupal eyes (scale bar = 5 μm) stained with Phalloidin illustrate the rescue of photoreceptor cell organization defects due to knockdown of *Cbp20* or *dlg1* upon overexpression of *Diap1*. (**F**) Larval eye discs (scale bar = 30 μm) stained with TUNEL show rescue of apoptosis phenotypes observed in flies with knockdown of *Cbp20* or *dlg1* and overexpression of *Diap1*, as well as enhanced apoptosis with overexpression of *Dronc*. (**G**) Box plot of TUNEL-positive cells in the larval eye discs of flies with knockdown of *Cbp20* or *dlg1* and overexpression of *Diap1* or *Dronc* is shown (n = 7–10, *p < 0.05, two-tailed Mann–Whitney test with Benjamini-Hochberg correction). All boxplots indicate median (center line), 25th and 75th percentiles (bounds of box), and minimum and maximum (whiskers), with red dotted lines representing the control median. A list of full genotypes for fly crosses used in these experiments is provided in **S2 File**. (PDF)

**S11 Fig. Phenotypic scores for interactions between homologs of 3q29 genes and known neurodevelopmental genes in the adult fly eye.** (**A-D**) Box plots of phenotypic scores for concomitant *GMR-GAL4* RNAi knockdown of fly homologs of 3q29 genes and neurodevelopmental genes, compared with recombined lines for individual homologs of 3q29 genes crossed with controls, are shown (n = 2–10, *p < 0.05, two-tailed Mann–Whitney test with Benjamini-Hochberg correction). All boxplots indicate median (center line), 25th and 75th percentiles (bounds of box), and minimum and maximum (whiskers), with red dotted lines representing the control median. A list of full genotypes for fly crosses used in these experiments is provided in **S2 File**. (PDF)

**S12 Fig. Quantification of 3q29 morpholino knockdown and apoptosis marker levels in *X. laevis* models.** (**A**) Electrophoretic gels show decreased expression of homologs of 3q29 genes due to morpholino (MO) knockdown at various concentrations in *X. laevis* embryos. Three replicates (uninjected and two MO concentrations) were performed for each morpholino, and band intensities were compared with expression of *ODC1* controls taken from the same cDNA samples and run on gels processed in parallel. (**B**) Quantification of expression for homologs of 3q29 genes at different MO concentrations, as measured by band intensity ratio to *ODC1* controls (n = 3 replicates, *p<0.05, two-tailed Welch's T-test with Benjamini-Hochberg correction). (**C**) Full images of western blots for quantification of cleaved caspase-3 levels in *X. laevis* embryos with MO knockdown of homologs of 3q29 genes. Two replicate experiments were performed, and the intensity of bands at 19kD and 17kD (green arrows), corresponding with cleaved caspase-3, were normalized to those for the β-actin loading controls. Embryos injected with control MO, uninjected embryos, and embryos treated with 30% EtOH as a positive control were included with the embryos injected with 3q29 MOs. (PDF)

**S13 Fig. Eye phenotypes observed with knockdown of homologs of 3q29 genes in *X. laevis* models.** (**A**) Representative eye images of stage 42 *X. laevis* tadpoles with MO knockdown of homologs of 3q29 genes (scale bar = 500 μm) show defects in eye size and morphology compared with the control (top). These defects were rescued with co-injection and overexpression of mRNA for homologs of 3q29 genes, as well as overexpression of the apoptosis inhibitor *xiap* for *ncbp2* (bottom). (**B**) Box plot of eye area in *X. laevis* models with knockdown of homologs of 3q29 genes, normalized to controls, is shown (n = 48–71, *p < 0.05, two-tailed Welch's T-

test with Benjamini-Hochberg correction). Models with *ncbp2* knockdown and *xiap* overexpression showed an increased eye size compared with *ncbp2* knockdown. (**C**) Box plot of eye area in *X. laevis* models with knockdown of homologs of 3q29 genes and overexpression of mRNA for homologs of 3q29 genes, normalized to controls, is shown (n = 56–63, *p < 0.05, two-tailed Welch's T-test with Benjamini-Hochberg correction). All boxplots indicate median (center line), 25th and 75th percentiles (bounds of box), and minimum and maximum (whiskers), with red dotted lines representing the control median. The data shown for the eye area experiments represent pooled results of three experimental batches, and were normalized to the respective controls from each batch.
(PDF)

**S14 Fig. Replication of *Drosophila* experimental results for individual and pairwise knockdown of homologs of 3q29 genes.** (**A**) Replication dataset for climbing ability of select homologs of 3q29 genes over ten days. We replicated the defects in climbing ability observed with pan-neuronal RNAi knockdown of *Cbp20* and *dlg1*, while climbing defects in flies with knockdown of *Fsn* flies were not replicated in the second experimental batch and were therefore excluded from the main dataset (**Fig 2B**). Data represented show mean ± standard deviation of 7–10 independent groups of 10 flies for each homolog. (**B**) Replication dataset for climbing ability of pairwise knockdown of homologs of 3q29 genes over ten days. We replicated the defects in climbing ability observed with pan-neuronal RNAi knockdown of *Cbp20/dlg1* and *Cbp20/Fsn* compared with recombined *Cbp20* knockdown crossed with control (**Fig 3F**). Data represented show mean ± standard deviation of 5 independent groups of 10 flies for each homolog. (**C**) Replication dataset for adult eye area in flies with *GMR-GAL4* RNAi knockdown of homologs of 3q29 genes (n = 10–14, *p < 0.05, two-tailed Mann–Whitney test with Benjamini-Hochberg correction). We replicated the decreased eye sizes in flies with knockdown of *Cbp20* and *CG8888*, while flies with knockdown of *dlg1* showed a non-significant (p = 0.154) increase in eye size (**Fig 2D**). (**D**) Confocal images for replication dataset larval eye discs (scale bar = 30 μm) stained with anti-pH3 (top) and TUNEL (bottom) illustrate cellular defects posterior to the morphogenetic furrow (white box) upon knockdown of select fly homologs of 3q29 genes (**Fig 2E**). (**E**) Replication dataset for pH3-positive cells in larval eye discs of flies with knockdown of homologs of 3q29 genes (n = 9–10, two-tailed Mann–Whitney test with Benjamini-Hochberg correction). As in the main dataset (**Fig 2F**), we observed no significant changes in cell proliferation for flies with knockdown of *Cbp20* and *dlg1*. (**F**) Replication dataset for TUNEL-positive cells in larval eye discs of flies with knockdown of homologs of 3q29 genes (n = 6–8, *p < 0.05, two-tailed Mann–Whitney test with Benjamini-Hochberg correction). We replicated the increased apoptosis phenotypes observed with knockdown of *Cbp20* and *dlg1* (**S4 Fig**). All boxplots indicate median (center line), 25th and 75th percentiles (bounds of box), and minimum and maximum (whiskers), with red dotted lines representing the control median. A list of full genotypes for fly crosses used in these experiments is provided in **S2 File**.
(PDF)

**S1 Table. *Drosophila* homologs of human 3q29 genes and expression of *Drosophila* homologs during development.** DIOPT version 7.1 [100] and reciprocal BLAST were used to identify fly homologs of genes within the 3q29 region; six genes did not have fly homologs. Expression levels of fly homologs of 3q29 genes were assessed using high-throughput expression data from FlyAtlas Anatomy microarray expression data [104] and modENCODE Anatomy RNA-Seq data [105] from FlyBase.
(PDF)

**S2 Table. qPCR primers and expression values for RNAi knockdown of fly homologs of 3q29 genes.** *Elav-GAL4* flies were crossed with RNAi lines of fly homologs of 3q29 genes at 25˚C, and 3–4 day old adult *Drosophila* heads were used to quantify the level of expression compared with *Elav-GAL4* controls. *Elav-GAL4;;UAS-Dicer2* flies crossed with *CG5359* flies showed overexpression of *tiptop* [103] and were therefore excluded from further experiments. A list of full genotypes for fly crosses used in these experiments is provided in **S2 File**, and statistics for these data are provided in **S5 File**.
(PDF)

**S3 Table. Comparison of animal model phenotypes with knockdown or knockout of homologs of 3q29 genes.** Blue shaded boxes indicate previously identified phenotypes for individual homologs of 3q29 genes, while "X" marks indicate recapitulated and novel phenotypes identified in our study. Gray-shaded boxes indicate that a homolog was not present in the model organism. Fly phenotypes were obtained from FlyBase [122], *X. laevis* phenotypes were obtained from Xenbase [120], and mouse knockout model phenotypes were obtained from the Mouse Genome Informatics database [123].
(PDF)

**S4 Table. Summary of scoring for phenotypic severity of axon targeting defects upon individual and pairwise knockdown of homologs of 3q29 genes.** Individual larval eye disc images were assigned mild, moderate or severe scores based on the severity of axon projection loss observed in each eye disc (see Methods). We found that the mild to moderate defects observed with knockdown of *Cbp20* were enhanced with concomitant knockdown of *dlg1* or *Fsn*, while *Diap1* overexpression partially rescued the defects observed with knockdown of *Cbp20* or *dlg1*. A list of full genotypes for fly crosses used in these experiments is provided in **S2 File**.
(PDF)

**S5 Table. Comparison of eye phenotypic scores for homologs of 3q29 genes and neurodevelopmental genes.** Table comparing *Flynotyper* scores for flies with *GMR-GAL4;UAS-Dicer2* RNAi knockdown of homologs of 3q29 genes (shaded in grey) with previously published scores for flies with *GMR-GAL4;UAS-Dicer2* RNAi knockdown of homologs of candidate neurodevelopmental genes [53].
(PDF)

**S6 Table. Analysis of defects in ommatidial cells with *GMR-GAL4* RNAi knockdown of fly homologs of 3q29 genes.** The number of "+" symbols displayed in the table indicate the severity of the observed cellular defects. Note that n = 4–16 pupal eye preparations were assessed for each RNAi line tested. A list of full genotypes for fly crosses used in these experiments is provided in **S2 File**.
(PDF)

**S7 Table. Screening for pairwise interactions among fly homologs of 3q29 genes.** "All interactions" indicates the number of pairwise crosses where at least one second-hit RNAi or mutant line showed enhancement of the single-hit phenotype, while "Validated" indicates the number of interactions which have two or more crosses with a second-hit RNAi or mutant line (if available) showing the same result. "Reciprocal cross" indicates the number of interactions with concordant results across pairs of reciprocal cross (i.e. *Cbp20*/*dlg1* vs. *dlg1*/*Cbp20*). These totals include crosses with the mutant line *Tsf2^{KG01571}*, as eye-specific RNAi knockdown of *Tsf2* was lethal, as well as flies heterozygous for *dlg1* RNAi and homozygous for *Cbp20* RNAi. Crosses with other RNAi or mutant lines for the same homolog (shaded in grey) are included as validation lines tested but were not counted as interactions. A list of full genotypes

for fly crosses used in these experiments is provided in S2 File.
(PDF)

**S8 Table. Analysis of defects in ommatidial cells with pairwise *GMR-GAL4* RNAi knockdown of fly homologs of 3q29 genes.** The number of "+" symbols displayed in the table indicate the severity of the observed cellular defects. Note that n = 4–16 pupal eye preparations were assessed for each interaction cross tested. A list of full genotypes for fly crosses used in these experiments is provided in S2 File.
(PDF)

**S9 Table. Screening for interactions between fly homologs of 3q29 genes and other known neurodevelopmental genes.** "All interactions" indicates the number of crosses where at least one second-hit RNAi line showed enhancement of the single-hit phenotype, while "Validated interactions" indicates the number of interactions which have two or more crosses with a second-hit RNAi or mutant line (if available) showing the same result. Results from two distinct fly homologs of *CHRNA7* that were crossed with homologs of 3q29 genes, *nAChRα6* and *nAChRα7*, were combined for the final number of interactions. Shaded interactions indicate pairwise crosses where the phenotypes observed with knockdown of the homolog for the neurodevelopmental gene by itself were suppressed with concomitant knockdown of homologs for 3q29 genes. The neurodevelopmental genes are annotated for cell cycle/apoptosis function (Gene Ontology terms GO:0007049 and GO:0006915) and association with microcephaly disorders [65]. A list of full genotypes for fly crosses used in these experiments is provided in S2 File.
(PDF)

**S10 Table. Developmental phenotypes observed in mouse models of the 3q29 deletion and individual homologs of 3q29 genes.** Comparison of mice with heterozygous deletion of the syntenic 3q29 region [14,15] with heterozygous knockout mouse models for *Dlg1* [14] and *Pak2* [72]. Blue shaded boxes indicate phenotypes observed in the knockout models, while gray-shaded boxes indicate a phenotype that was not tested in the knockout model. Neither *Dlg1*[+/-] nor *Pak2*[+/-] knockout mice recapitulate the body and brain weight, spatial learning and memory, or acoustic startle defects observed in the deletion mouse models.
(PDF)

**S11 Table. Summary of apoptosis function enrichment among candidate neurodevelopmental genes.** This table shows the number of candidate autism, intellectual disability and schizophrenia genes annotated for apoptosis function. The minimum, mean and maximum numbers of apoptosis genes in 100,000 simulated sets of candidate genes are shown, along with the percentiles and empirical p-values of the observed overlap with apoptosis genes for each simulation.
(PDF)

**S12 Table. Morpholinos used for *X. laevis* experiments.**
(PDF)

**S13 Table. qPCR primers used for *X. laevis* experiments.**
(PDF)

**S1 File. Pathogenicity metrics, mutations in disease cohorts, and biological functions of 3q29 genes.** 3q29 genes with Residual Variation Intolerance Scores (RVIS) <20[th] percentile [124] or probability of Loss-of-function Intolerant (pLI) scores >0.9 [125] are considered to be potentially pathogenic in humans and are shaded in gray. Mutations within 3q29 genes

identified in disease cohorts were curated from three databases: denovo-db v.1.6.1 [75], Gene-Book database (http://atgu.mgh.harvard.edu/~spurcell/genebook/genebook.cgi), and SFARI Gene [77]. Molecular functions for 3q29 genes were derived from RefSeq, UniProtKB and Gene Ontology (GO) individual gene summaries [126–128], and GO-SLIM terms for human genes and fly homologs were curated from PantherDB [101]. Annotations for cell cycle/apoptosis and neuronal function were derived from GO Biological Process annotations for each gene.
(XLSX)

**S2 File. List of fly stocks and full genotypes for all crosses tested.** This file lists the stock lines, stock center, and genotypes for primary and validation lines for fly homologs of 3q29 genes as well as neurodevelopmental and apoptosis genes outside of the 3q29 region. Full genotypes for the generated recombined lines as well as all individual and pairwise crosses tested in the manuscript are also listed in the file. BDSC: Bloomington *Drosophila* Stock Center; VDRC: Vienna *Drosophila* Resource Centre.
(XLSX)

**S3 File. Transcriptome analysis of flies with knockdown of homologs of 3q29 genes.** This file lists all differentially expressed genes from RNA sequencing of flies with *Elav-GAL4* RNAi knockdown of homologs of 3q29 genes, as defined by log-fold change >1 or < -1 and false discovery rate (FDR) <0.05 (Benjamini-Hochberg correction). Human homologs identified using DIOPT are included for each differentially-expressed fly gene. The file also includes enriched Gene Ontology (GO) terms (p<0.05, Fisher's Exact test with Benjamini-Hochberg correction) for each set of differentially-expressed fly genes, as well as lists of GO terms enriched among their corresponding human homologs.
(XLSX)

**S4 File. List of candidate neurodevelopmental genes with apoptosis function.** This file lists 525 candidate neurodevelopmental genes that are annotated for apoptosis GO terms, including their membership within pathogenic CNV regions.
(XLSX)

**S5 File. Statistical analysis of experimental data.** This file shows all statistical information (sample size, mean/median/standard deviation of datasets, Shapiro-Wilk test statistics for normality, controls used, test statistics, p-values, confidence intervals, and Benjamini-Hochberg FDR corrections) for all data presented in the main and supplemental figures. Statistical information for ANOVA tests includes factors, degrees of freedom, test statistics, and post-hoc pairwise t-tests with Benjamini-Hochberg correction.
(XLSX)

**S1 Video. Climbing ability of flies with knockdown of individual homologs of 3q29 genes.** This video shows the climbing ability of *Elav-GAL4* control, *Cbp20* and *dlg1* individual RNAi knockdown flies at day 10 of the climbing ability experiments.
(MP4)

**S2 Video. Climbing ability of flies with pairwise knockdowns of homologs of 3q29 genes.** This video shows the climbing ability of *Cbp20/dlg1* and *Cbp20/Fsn* pairwise *Elav-GAL4* RNAi knockdown flies at day 10 of the climbing ability experiments.
(MP4)

## Acknowledgments

We thank A. Krishnan for assistance with the brain-specific gene interaction network analysis, J. Tiber for technical assistance with the *X. laevis* experiments, and V. Faundez for useful discussions and critical reading of the manuscript.

## Author Contributions

**Conceptualization:** Mayanglambam Dhruba Singh, Matthew Jensen, Santhosh Girirajan.

**Formal analysis:** Mayanglambam Dhruba Singh, Matthew Jensen, Micaela Lasser.

**Funding acquisition:** Laura Anne Lowery, Santhosh Girirajan.

**Investigation:** Mayanglambam Dhruba Singh, Matthew Jensen, Micaela Lasser, Emily Huber, Tanzeen Yusuff, Lucilla Pizzo, Brian Lifschutz, Inshya Desai, Alexis Kubina, Sneha Yennawar, Sydney Kim, Janani Iyer.

**Project administration:** Santhosh Girirajan.

**Supervision:** Diego E. Rincon-Limas, Laura Anne Lowery, Santhosh Girirajan.

**Visualization:** Mayanglambam Dhruba Singh, Matthew Jensen.

**Writing – original draft:** Mayanglambam Dhruba Singh, Matthew Jensen, Santhosh Girirajan.

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
