## [Decision Letter · Decision Letter 0]

20 Nov 2019

Dear Dr Girirajan,

Thank you very much for submitting your Research Article entitled 'NCBP2 modulates neurodevelopmental defects of the 3q29 deletion in Drosophila and X. laevis models' to PLOS Genetics. Your manuscript was fully evaluated at the editorial level and by independent peer reviewers. The reviewers appreciated the attention to an important topic but identified some aspects of the manuscript that should be improved.

We therefore ask you to modify the manuscript according to the review recommendations before we can consider your manuscript for acceptance. Your revisions should address the specific points made by each reviewer.

[LINK]

Yours sincerely,

Gregory S. Barsh

Editor-in-Chief

PLOS Genetics

Gregory Copenhaver

Editor-in-Chief

PLOS Genetics

Reviewer's Responses to Questions

**Comments to the Authors:**

Reviewer #1: The paper entitled NCBP2 modulates neurodevelopmental defects of the 3q29 deletion in Drosophila and X. laevis models focuses on a series of putative gene interactions identified by examination of the 3q29 deletion, which has been associated with multiple neurodevelopmental disorders. The authors suggest that the loss of multiple genes from within the deletion can be synergistically deleterious, resulting in more significant phenotypes in 44 of the tested pairwise knockout animals. Of particular note was the gene NCBP2, which appeared to enhance the phenotypes of other 3q29 genes when both were lost simultaneously. The authors also noted that many of the observed phenotypes appeared to be the result of defects in cellular organization and apoptosis, and found that the defects could be, at least partially, rescued by the expression of apoptosis inhibitors. In support of these claims, the authors provided the following evidence:

1. Knock-down of 3q29 homologues (when available, using the daughterless Gal4 line) in Drosophila results in global developmental defects, including lethality.

2. Knock-down of 3q29 homologues in the Drosophila eye (using GMR-Gal4) resulted in cellular organization defects consistent with other known neurodevelopmental genes

3. Pairwise knockdown of 3q29 homologues results in more extreme eye phenotypes, as identified by flynotyper software.

4. A subset of the 3q29 genes appear to be highly interconnected, based on mined gene interaction data.

5. Pairwise knockdown of the NCBP2 homologue Cbp20 and other 3q29 homologues results in enhanced apoptosis.

6. Pairwise knockdown of Cbp20 and known neurodevelopmental genes increased the number and severity of ommitidial necrotic patches.

7. Knock down of 3q29 genes using morpholono oligonucleotides in a Xenopus model was sufficient to cause developmental defects on the injected side of the animal.

The work presented as part of this manuscript is an interesting initial exploration of the genes contained within the 3q29 deletion. By examining the loss of both single genes and pairs from within the 3q29 locus, the authors provide evidence that a small subset of genes and their interactions may impact specific biological processes relevant to nervous system development. In particular, it would appear that NCBP2 is of particular importance, as loss of the fly homologue (Cbp20) resulted in more severe eye phenotypes when it was knocked out in a pairwise manner with 11 of the 12 other 3q29 homologues. This form of functional validation is of particular importance, because, according to the authors NCBP2 is not predicted to be pathogenic and does not have identified deleterious mutations. The authors also proposed that genes within the 3q29 locus could alter multiple cellular processes, including control of the cell cycle and apoptosis, which they then demonstrate in Drosophila and Xenopus models. More broadly, the work presented in this manuscript provides indirect evidence in support of the idea that abnormal regulation of apoptosis may be a key contributor to neurological disorders such as autism and schizophrenia. While the authors do not identify a specific mechanism as a potential culprit for any of the 3q29 associated neuropsychatric disorders, they do identify several potential interaction partners, and highlight some of the more essential genes within the 3q29 locus.

While I do have some concerns (denoted below) with some of the data provided, I believe this manuscript to be a generally positive contribution to our understanding of the importance of the 3q29 locus.

Requested Revisions / Experiments / Explanations:

1. We are concerned about the use of the terms “pathogenicity” and “causative” in several places throughout the manuscript. The American College for Medical Genetics defines criteria for determing the pathogenicity of specific sequence variants for medical genetics. The deletions being discussed in this paper are often already considered pathogenic in patients without the data presented by the authors here. The authors seem to use the term pathogenicity when they mean the potential for underlying mechanism in gene interactions, but fundamentally this paper does not address a disease mechanism therefore this is not demonstrated pathogenicity. We would request that they remove entirely any use or reference of their work in flies using pairwise crosses to denote pathogenicity defined in humans. The first example includes within the abstract “These cellular and neuronal defects were rescued…suggesting that apopotosis is one of several potential biological mechanisms for pathogenicity of the deletion.” AND “Overall our study suggests that NCPB2-mediated genetic interactions disrupt apoptosis and cell cycle mechanisms during development, contributing to the neurodevelopmental features of the 3q29 deletion.” These statement make too many assumptions about the shared processes between flies and humans. The authors need to scale back these expansive clinical conclusions and simply describe their data. They should remove “pathogenicity” from every instance in this manuscript.

2. Figure 2C: The phenotype scores reported in figure 2C for dlg1 and Cbp20 do not reflect what is reported in other figures. For example, 2C reports an average score of ~40 for Cpb20, while it has an average score of ~30 in figure 3B. This is concerning, and calls into question the inter-experiment accuracy of flynotyper.

3. Figure 2E / Line 1319-1320: “To account for reduced DLG expression in pupal eyes with knockdown of dlg1, images were taken at a higher intensity than control images”. All images should be taken with a standardized protocol to allow for accurate comparison.

4. Line 231-232: The wording here is somewhat confusing in relation to the figure 2E. Does this not cover the entire area reported in the figure? If not, why was the region “adjacent” to the morphogenic furrow chosen as opposed to the entire developing eye?

5. Line 233-234: Please report the p-value for the PIG-Z and dlg1 “trends towards increased pH3-positive cells”.

6. Line 256-258: Please report the nature of the enhancement in phenotypic severity in the eye, such as the distance or angle OD score.

7. Figure S4: How are the axon phenotypes scored? What is the difference between mild/moderate/severe?

8. Figure S10A: The panel displaying the cell organization of animals overexpressing Dronc is missing.

9. Figure 7B / Line 398-390: Please provide a quantification of the necrotic patches shown in addition to the representative figures.

10. Figure 8D: The western blot for apoptosis markers appears to have multiple caspase-3 bands for the fbxo45 knockdown and fbxo45/ncbp2 knockdown, which are absent in other lanes. Is there an explanation of these bands? Is it possible there could be an error the measurement of band intensity due to their presence?

Reviewer #2: Your manuscript is very interesting both in terms of genetic screens and for the identification of candidate genes for the 3q29 region. Nevertheless I have a few questions regarding the validation of the down-regulation of the 3q29 homologs genes.

You only showed the s2 table for Elav-Gal4 but did you perform the validation of the levels of expression using other GAL4 drivers in drosophila or with morpholinos in Xenopus L? Can you show some validation experiments and in particular have you done it while performing gene interaction study? Were you able to find your gene of interest down-expressed using the RNAseq experiments in fly? It is of particular importance because as you stated in mouse heterozygous animals for Dlg1 and Pak2 do no show any phenotype.

**Have all data underlying the figures and results presented in the manuscript been provided?**

Reviewer #1: Yes

Reviewer #2: Yes

PLOS authors have the option to publish the peer review history of their article (what does this mean?). If published, this will include your full peer review and any attached files.

Reviewer #1: No

Reviewer #2: Yes: Yann Hérault

---

## [Editor Report · Decision Letter 1]

30 Dec 2019

Dear Santhosh,

Happy holidays!

We are pleased to inform you that your manuscript entitled "NCBP2 modulates neurodevelopmental defects of the 3q29 deletion in Drosophila and X. laevis models" has been editorially accepted for publication in PLOS Genetics. Congratulations!

Yours sincerely,

Gregory S. Barsh

Editor-in-Chief

PLOS Genetics

Gregory Copenhaver

Editor-in-Chief

PLOS Genetics

Comments from the reviewers (if applicable):

**Data Deposition**

http://datadryad.org/submit?journalID=pgenetics&manu=PGENETICS-D-19-01766R1

**Press Queries**

---

## [Editor Report · Acceptance letter]

4 Feb 2020

PGENETICS-D-19-01766R1 

NCBP2 modulates neurodevelopmental defects of the 3q29 deletion in Drosophila and Xenopus. laevis models 

Dear Dr Girirajan, 

We are pleased to inform you that your manuscript entitled "NCBP2 modulates neurodevelopmental defects of the 3q29 deletion in Drosophila and Xenopus. laevis models" has been formally accepted for publication in PLOS Genetics! Your manuscript is now with our production department and you will be notified of the publication date in due course.

With kind regards,

Matt Lyles

PLOS Genetics

On behalf of:
